# 🐰 RaBiT: Residual-Aware Binarization Training for Accurate and Efficient LLMs

## Abstract

Efficient deployment of large language models (LLMs) requires extreme quantization, forcing a critical trade-off between low-bit efficiency and performance. Residual binarization promises hardware-friendly, matmul-free inference by stacking binary ($\pm 1$) layers, but is plagued by pathological feature **co-adaptation**. We identify a key failure mode, which we term **inter-path adaptation**: during Quantization-Aware Training (QAT), parallel residual binary paths learn redundant features, degrading the error-compensation structure and crippling the model's expressive capacity. While prior work relies on heuristic workarounds (*e.g.,* path freezing) that limit model capacity, we propose **RaBiT**, a novel quantization framework that resolves co-adaptation by algorithmically enforcing a residual hierarchy. Its core mechanism sequentially derives each binary path from a single shared full-precision weight, ensuring each path corrects its predecessor's error. This process is stabilized by a robust initialization that prioritizes functional preservation over mere weight approximation. RaBiT redefines the 2-bit accuracy-efficiency frontier: it achieves state-of-the-art performance, rivals even hardware-intensive Vector Quantization (VQ) methods, and delivers a **4.49× inference speed-up** over full-precision models on an RTX 4090.

## 1 Introduction

The massive scale of large language models (LLMs) makes model compression essential for their efficient deployment. While 4-bit quantization methods (Frantar et al., 2023; Lin et al., 2024) have emerged as a successful industry standard (Kwon et al., 2023; Zheng et al., 2024), the relentless pursuit of greater efficiency is pushing the research frontier toward the extreme 2-bit regime. This push toward lower bit compression, however, introduces a critical architectural trade-off that defines the current landscape.

At this frontier, two dominant strategies present a stark choice between accuracy and hardware efficiency. On one hand, Vector Quantization (VQ) methods achieve high accuracy but often introduce hardware overhead from lookup tables or complex rotations (Tseng et al., 2024a;b; Egiazarian et al., 2024). On the other hand, residual binarization—stacking multiple binary layers—offers exceptional, matmul-free efficiency. Yet, this highly efficient approach has consistently struggled to maintain performance, hampered by fundamental training challenges that have prevented it from realizing its full potential (Bulat et al., 2024; Wang et al., 2024; Tran & Nguyen, 2025).

The core promise of a residual architecture—that subsequent paths compensate for the errors of preceding ones—is fundamentally undermined by feature **co-adaptation** (Hinton et al., 2012), a pathological training dynamic where parallel components learn redundant features. In residual binarization, we identify a critical manifestation of this phenomenon, which we term **inter-path adaptation**. During standard Quantization-Aware Training (QAT) (Bengio et al., 2013; Hubara et al., 2018), the structurally agnostic global gradient is applied to all paths simultaneously. This forces them to learn redundant features in a race to minimize the global objective, overriding their intended compensatory roles. The result is a breakdown of the residual hierarchy that severely limits the model's expressive power.

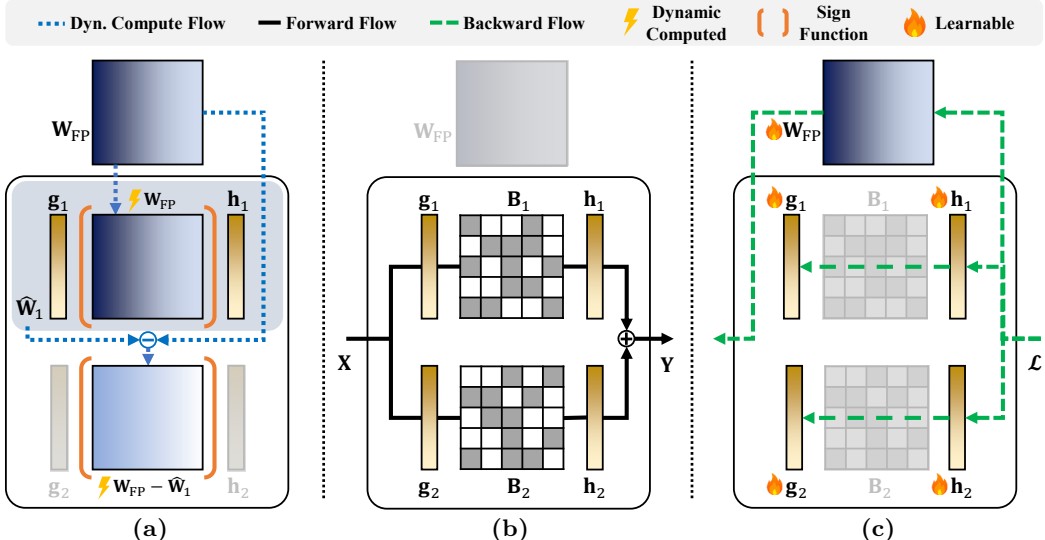

Figure 1: **Overview of the RaBiT training framework. (a) Dynamic Compute Process:** During training, binary paths are dynamically derived from a shared weight $\mathbf{W}_{\text{FP}}$ to enforce a residual hierarchy. **(b) Forward Pass:** For inference, these paths execute in parallel for matmul-free efficiency. **(c) Backward Pass:** Gradients from the loss $\mathcal{L}$ update both the learnable scales $(\mathbf{g}_i, \mathbf{h}_i)$ and the shared $\mathbf{W}_{\text{FP}}$.

Prior attempts to mitigate co-adaptation have relied on heuristic workarounds, such as freezing paths (Bulat et al., 2024; Tran & Nguyen, 2025), which limit the model's capacity to find an optimal joint solution. To address this, we propose **Residual-Aware Binarization Training (RaBiT)**, a QAT framework that resolves inter-path adaptation by design, as depicted in Figure 1. Instead of using independent latent weights, RaBiT maintains a single shared full-precision weight from which binary paths are sequentially derived on-the-fly, guided by learnable scales. This algorithmically enforces the residual hierarchy, training each path to correct its predecessor's error. Combined with a robust, function-aware initialization, RaBiT achieves state-of-the-art accuracy while delivering a 4.49× inference speed-up and halving the training memory footprint.

Our contributions can be summarized as follows:

- We identify and analyze **inter-path adaptation**, a critical manifestation of feature co-adaptation in residual binarization, where the intended error-compensation structure breaks down during Standard QAT as parallel paths become functionally redundant.
- We propose **RaBiT**, a novel QAT framework that resolves inter-path adaptation by enforcing **residual coupling** on-the-fly. The mechanism inherently **halves the training memory footprint** and is stabilized by a robust **function-aware initialization** strategy to tame the unstable dynamics of extreme QAT.
- We demonstrate that RaBiT achieves **state-of-the-art accuracy** at 2-bit precision, delivering a 4.49× inference speed-up while maintaining competitive performance against hardware-intensive VQ methods through matmul-free operations.

## 2 RELATED WORKS

**The Shift to QAT in Extreme Quantization.** Post-Training Quantization (PTQ) methods, such as GPTQ (Frantar et al., 2023) and AWQ (Lin et al., 2024), have proven highly successful for compressing large language models to 3- or 4-bit precision by focusing on weight approximation. However, these techniques face a steep performance cliff at lower bit-widths (*e.g.,* 2-bit) (Wang et al., 2023; Huang et al., 2024b; Li et al., 2025), as the information loss from coarse quantization becomes too severe to overcome by simply minimizing weight reconstruction error. Consequently, the research community is shifting from approximating weights to preserving the model's overall **functionality** (Liu et al.,

2025) through Quantization-Aware Training (QAT) (Hubara et al., 2018; Krishnamoorthi, 2018). QAT integrates the simulation of low-precision arithmetic into the fine-tuning process, allowing the model to adapt its parameters to the constraints of the target bit-width. While QAT is challenging due to the non-differentiable nature of quantization—typically addressed with the Straight-Through Estimator (STE) (Bengio et al., 2013)—modern frameworks for binary models have found stability by maintaining a latent full-precision weight for training and updating it via a surrogate gradient (Wang et al., 2023; Xu et al., 2024; Jo et al., 2024; Lee et al., 2025). Our work builds on this robust method to address the unique challenges of residual binary architectures.

**Co-adaptation in Residual Binary Architectures.** To enhance the limited expressive capacity of a single low-bit layer, residual binarization stacks multiple low-bit paths ($\mathbf{W} \approx \sum_i \tilde{\mathbf{W}}_i$) to achieve higher precision while retaining matmul-free efficiency (Wang et al., 2024). However, this parallel architecture is highly susceptible to **feature co-adaptation** (Hinton et al., 2012), a training pathology where components learn redundant features. This phenomenon, which spurred the development of regularization techniques like Dropout (Srivastava et al., 2014).

In the context of residual binarization, we identify a critical form of feature co-adaptation, termed **inter-path adaptation**, where a shared QAT gradient forces parallel paths to learn redundant features, undermining their error-compensation hierarchy. While prior work relied on suboptimal heuristics like path freezing (Bulat et al., 2024; Tran & Nguyen, 2025) that preclude finding a joint optimal solution, RaBiT resolves this core challenge by design, enabling true joint optimization while algorithmically enforcing the hierarchy.

## 3 Motivation

The goal of Quantization-Aware Training (QAT) is to make a quantized student model, $\mathbf{Y}_s$, functionally mimic its full-precision teacher, $\mathbf{Y}_t$. This is typically achieved by optimizing an objective that combines the final task loss with an intermediate knowledge distillation loss, often formulated as the mean squared error (MSE) (Hinton et al., 2015; Liu et al., 2024). While our full training objective also includes the final KL divergence-based task loss, we initially focus our analysis on the MSE component for its analytical tractability. We formally prove in Appendix A.2 that this error-decomposition logic and the optimality of RaBiT's residual coupling extend rigorously to the KL divergence objective. The additive structure of the MSE provides a clear window into how parallel paths interact. In a 2-bit residual architecture, the MSE between the teacher output $y_t$ and the student output $y_s = y_1 + y_2$ can be decomposed. Using the Pearson correlation coefficient,[1] this decomposition is:

$$\text{MSE}(y_t, y_s) = \underbrace{(\mathbb{E}[y_t^2] + \mathbb{E}[y_1^2] + \mathbb{E}[y_2^2] - 2\mathbb{E}[y_t y_s])}_{C'} + \underbrace{2\sigma_1 \sigma_2}_{PathAmp.} \cdot \underbrace{\text{Corr}(y_1, y_2)}_{PathCorr.}$$

where $C'$ represents the sum of correlation-independent error terms. This reveals a core principle: to minimize the MSE, the paths must be strongly **negatively correlated**. A negative correlation transforms the interaction term into a substantial **bonus** that actively reduces the total loss, signifying effective error-cancellation.

However, Standard QAT structurally fails to achieve this. The shared global gradient induces **inter-path adaptation**, forcing both paths to learn redundant features instead of their intended compensatory roles (see Appendix A.1). To provide a concrete analysis, we decompose the MSE loss for representative layers of Llama2-7B, selected to show the characteristics across the early, middle, and late stages of the network, in Table 1.

The analysis in Table 1 is definitive. Across early, middle, and late stages of the network, the base error term $C'$ and the path amplitude $2\sigma_1 \sigma_2$ remain comparable between both methods. The critical difference lies in the correlation. Standard QAT yields a correlation close to zero, resulting in a negligible interaction term that fails to meaningfully reduce the

---

[1]The relationship $\mathbb{E}[y_1 y_2] \approx \sigma_1 \sigma_2 Corr(y_1, y_2)$ relies on a zero-mean assumption for the path outputs. We empirically verify this, finding the omitted $\mathbb{E}[y_1]\mathbb{E}[y_2]$ term is less than 1% of the covariance term and thus negligible.

Table 1: **Detailed Decomposition of MSE Loss across Representative Layers of Llama2-7B.** The table breaks down the total MSE into its core components. While the base error ($C'$) and path amplitude ($2\sigma_1\sigma_2$) are comparable, RaBiT consistently generates a strong negative correlation, creating a significant loss-reducing **Bonus**. In contrast, Standard QAT's weak correlation provides a negligible benefit, demonstrating RaBiT's structural advantage in error correction.

| Layer | Method | Base Error ($C'$) | Path Amp. ($2\sigma_1\sigma_2$) | Path Corr. (Corr) | Covariance (Amp. × Corr) | Total MSE ($C'$ + Cov.) |
|---|---|---|---|---|---|---|
| Layer 5 (Early) | Standard QAT | 0.0019 | 0.0030 | -0.0752 | -0.0002 | 0.0017 |
| | **RaBiT (Ours)** | 0.0023 | 0.0028 | **-0.4961** | **-0.0014** | **0.0009** |
| Layer 15 (Mid) | Standard QAT | 0.0182 | 0.0214 | -0.1240 | -0.0026 | 0.0156 |
| | **RaBiT (Ours)** | 0.0163 | 0.0200 | **-0.3418** | **-0.0068** | **0.0094** |
| Layer 25 (Late) | Standard QAT | 0.0575 | 0.0728 | -0.1279 | -0.0093 | 0.0482 |
| | **RaBiT (Ours)** | 0.0609 | 0.0801 | **-0.3535** | **-0.0283** | **0.0327** |

total error. In stark contrast, **RaBiT** structurally enforces a strong negative correlation (*e.g.,*-0.50 in layer 5) (see Appendix A.1). This transforms the interaction term into a significant loss-reducing bonus, systematically lowering the total MSE (see Appendix A.7). This principled enforcement of anti-correlation creates a more stable optimization landscape, leading to better generalization and superior performance.

## 4 METHOD

We introduce **RaBiT**, a novel QAT framework that prevents interference between the parallel paths of stacked binary architectures. To achieve this, RaBiT enforces a clear error-correction role for each path using a novel **coupled training** loop, and stabilizes the process with a **function-aware initialization** strategy. An overview of the RaBiT training framework is illustrated in Figure 1.

### 4.1 THE RESIDUAL BINARIZATION ARCHITECTURE

To achieve low-bit precision (*e.g.,* 2-bit) while maximizing computational efficiency, we adopt a residual architecture built upon highly efficient binary building blocks.

**Binary Building Blocks.** The fundamental component is the dual-scale binarization framework. We define the approximation of a weight matrix $\hat{\mathbf{W}}$ using a notation that highlights the underlying element-wise scaling operations:

$$\hat{\mathbf{W}} = \mathbf{g} \odot \mathbf{B} \odot \mathbf{h}. \tag{1}$$

Here, the expression denotes an operation where each element of the resulting matrix, $(\hat{\mathbf{W}})_{ij}$, is computed as $g_i B_{ij} h_j$. $\mathbf{B} \in \{-1, +1\}^{d_{\text{out}} \times d_{\text{in}}}$ is the binary core matrix, and $\mathbf{g} \in \mathbb{R}^{d_{\text{out}}}$, $\mathbf{h} \in \mathbb{R}^{d_{\text{in}}}$ are full-precision, per-channel scaling vectors. The key advantage of this formulation is its matmul-free efficiency. For an input vector $\mathbf{x} \in \mathbb{R}^{d_{\text{in}}}$, the forward operation computes the output vector $\mathbf{y} \in \mathbb{R}^{d_{\text{out}}}$ as $\mathbf{y} = \mathbf{g} \odot (\mathbf{B}(\mathbf{h} \odot \mathbf{x}))$, which can be implemented using only additions and subtractions, eliminating costly multiplications.

**Multi-bit Approximation via Stacking.** To enhance representational capacity (*e.g.,* to 2-bit) while retaining this efficiency, we stack $k = 2$ binary paths in parallel. The effective weight is the sum of two binarized terms:

$$\hat{\mathbf{W}}^{(k)} = \sum_{i=1}^{k} \hat{\mathbf{W}}_i = \sum_{i=1}^{k} \mathbf{g}_i \odot \mathbf{B}_i \odot \mathbf{h}_i. \tag{2}$$

This architecture preserves the underlying matmul-free execution, as the forward pass simply accumulates the outputs from each path.

## 4.2 Coupled Training for Co-Adaptation Mitigation

To address inter-path adaptation, RaBiT abandons the standard approach of training independent latent weights for each binary path. Instead, it maintains a **single shared full-precision (FP) weight** $\mathbf{W}_{\text{FP}}$ that serves as the anchor for the entire residual structure.

**The Coupled Forward Pass.** The core of RaBiT lies in its dynamic forward pass. For a 2-bit architecture with $k = 2$ paths, the binary core matrices, $\mathbf{B}_1$ and $\mathbf{B}_2$, are not stored but re-calculated during every forward pass from the shared weight $\mathbf{W}_{\text{FP}}$ (Figure 1a). This process algorithmically enforces the error-compensation hierarchy. Unlike the dynamically derived binary cores, the scaling vectors $\{\mathbf{g}_i, \mathbf{h}_i\}$ are independent, learnable parameters that capture the magnitude of each path. The derivation proceeds as follows:

1. **Path 1 Derivation**: The first binary core, $\mathbf{B}_1$, is determined by directly binarizing the shared weight: $\mathbf{B}_1 = \text{sign}(\mathbf{W}_{\text{FP}})$. This binary core is then combined with its corresponding learnable scaling vectors, $\mathbf{g}_1$ and $\mathbf{h}_1$, to reconstruct the first-path approximation, $\hat{\mathbf{W}}_1 = \mathbf{g}_1 \odot \mathbf{B}_1 \odot \mathbf{h}_1$.

2. **Residual Calculation**: The residual error, $\mathbf{R}_1$, is calculated by subtracting the *reconstructed* first path from the shared weight: $\mathbf{R}_1 = \mathbf{W}_{\text{FP}} - \hat{\mathbf{W}}_1$.

3. **Path 2 Derivation**: The second binary core, $\mathbf{B}_2$, is then determined by binarizing this freshly computed residual error: $\mathbf{B}_2 = \text{sign}(\mathbf{R}_1)$. The final effective weight used in the forward pass is the sum of the two reconstructed paths: $\hat{\mathbf{W}}^{(2)} = \hat{\mathbf{W}}_1 + (\mathbf{g}_2 \odot \mathbf{B}_2 \odot \mathbf{h}_2)$.

A key design choice is to derive only the **binary cores** $\mathbf{B}_i := \text{sign}(\mathbf{R}_{i-1})$ dynamically, while treating the **scaling vectors** $\{\mathbf{g}_i, \mathbf{h}_i\}$ as independent, learnable parameters. This separation of roles is crucial for both computational efficiency and training stability. Re-calculating optimal scales for the residual at every forward pass—*e.g.,* via Singular Value Decomposition (SVD)—would be prohibitively expensive. Consequently, by maintaining them as learnable parameters, the optimizer can leverage state accumulation (*e.g.,* momentum) to robustly fine-tune the well-initialized values (Section 4.3). This data-adaptive tuning is vital for training stability and allows the error-compensation hierarchy to function effectively by learning the optimal magnitude for each path.

**Backward Pass and Parameter Updates.** The backward pass is designed for stability and effectiveness. The gradient from the loss $\mathcal{L}$ flows back to update both the independent, learnable scaling vectors $\{\mathbf{g}_i, \mathbf{h}_i\}$ and the single shared weight $\mathbf{W}_{\text{FP}}$, as shown in Figure 1c.

- **Gradient for Learnable Scales**: The scaling vectors $\{\mathbf{g}_i, \mathbf{h}_i\}$ are treated as standard learnable parameters and receive their own gradients via the chain rule. For a mini-batch of size $B$, the gradients are accumulated over each sample, treating the dynamic binary cores ($\mathbf{B}_i$) as constants during this calculation:

$$\nabla_{\mathbf{g}_i} = \sum_{b=1}^{B} \Delta_b \odot \left(\mathbf{B}_i \left(\mathbf{h}_i \odot \mathbf{X}_b\right)\right), \quad \nabla_{\mathbf{h}_i} = \sum_{b=1}^{B} \left(\mathbf{B}_i^\top \left(\Delta_b \odot \mathbf{g}_i\right)\right) \odot \mathbf{X}_b. \tag{3}$$

  Here, $b$ is the sample index within the mini-batch, $\mathbf{X}_b$ is the input vector for that sample, and $\Delta_b = (\partial\mathcal{L}/\partial\mathbf{Y}_b)$ is the upstream gradient from the layer's output $\mathbf{Y}_b$ for that sample.

- **Gradient for the Shared Weight**: To update the single shared weight $\mathbf{W}_{\text{FP}}$, RaBiT uses an **effective-weight gradient**. This acts as a Straight-Through Estimator (STE) for the *entire coupled derivation process*. The gradient is computed with respect to the final effective weight $\hat{\mathbf{W}}^{(k)} = \sum_i \hat{\mathbf{W}}_i$ and is passed back directly to update $\mathbf{W}_{\text{FP}}$:

$$\nabla_{\mathbf{W}_{\text{FP}}} \approx \nabla_{\hat{\mathbf{W}}^{(k)}} \mathcal{L} = \left(\partial\mathcal{L}/\partial\mathbf{Y}\right)^\top \mathbf{X}. \tag{4}$$

In this context, $\mathcal{L}$ is the task loss, while $\mathbf{X}$ and $\mathbf{Y}$ represent the full input and output matrices for the mini-batch. This update completes the training loop: by recomputing the binary paths from the updated $\mathbf{W}_{\text{FP}}$ at every step, RaBiT continuously forces each path to correct the latest residual error, which preserves the overall error-compensation hierarchy.

For inference, the final binary cores $\{\mathbf{B}_i\}$ are derived from the trained $\mathbf{W}_{\mathrm{FP}}$ and then frozen, while the shared weight $\mathbf{W}_{\mathrm{FP}}$ is discarded. The resulting architecture is highly efficient, as the independent binary paths execute in a fully parallel, matmul-free manner. Crucially, this single-weight design also provides a key training advantage: by halving the latent parameters, it reduces the memory required for optimizer states by 50%, a major bottleneck in LLM fine-tuning. The complete training step is detailed in Algorithm 2.

### 4.3 STABLE INITIALIZATION FOR FUNCTIONAL PRESERVATION

QAT in the 2-bit regime is extremely sensitive to the initial quantization error. To mitigate this, we propose a two-stage initialization process that prioritizes preserving model *functionality* over merely approximating weight values.

**1. Iterative Residual SVID.** The core of our initialization is to find a set of binary paths that jointly approximate a target weight matrix. A standard greedy decomposition is suboptimal because the choice for the first path irreversibly biases all subsequent paths. To find a better joint solution, we propose **Iterative Residual Sign-Value-Independent Decomposition (SVID)**, a Gauss-Seidel style iteration that allows the paths to co-adapt. The process iteratively refines the scales $\{\mathbf{g}_i, \mathbf{h}_i\}$ and binary cores $\{\mathbf{B}_i\}$ for each path $i = 1, \ldots, k$ over $t = 1, \ldots, T$ iterations as follows:

$$\begin{cases} \mathbf{R}_i^{(t)} & := \mathbf{W}_{\mathrm{FP}} - \left( \sum_{j<i} \hat{\mathbf{W}}_j^{(t)} + \sum_{j>i} \hat{\mathbf{W}}_j^{(t-1)} \right), \\ \mathbf{B}_i^{(t)}, \mathbf{g}_i^{(t)}, \mathbf{h}_i^{(t)} & := \mathrm{SVID}(\mathbf{R}_i^{(t)}), \\ \hat{\mathbf{W}}_i^{(t)} & := \mathbf{g}_i^{(t)} \odot \mathbf{B}_i^{(t)} \odot \mathbf{h}_i^{(t)}. \end{cases} \quad (5)$$

Here, SVID($\cdot$) (Xu et al., 2024) finds optimal per-channel scales by separating the signs and performing a rank-1 SVD approximation on the magnitudes. Note that this iterative process is not directly applied to the raw weights $\mathbf{W}_{\mathrm{FP}}$, but to a preconditioned target matrix $\mathbf{W}'$, which we define next.

**2. I/O Channel Importance-Scaled Preconditioning.** To ensure our iterative decomposition focuses on the most functionally critical components of the weight matrix, we do not apply it to the raw weights $\mathbf{W}_{\mathrm{FP}}$. Instead, inspired by recent work on preserving functional saliency (Boža & Hradiš, 2025), we first **precondition** the matrix to create the target $\mathbf{W}'$. Using a small calibration dataset, we compute input activation magnitudes ($\mathbf{s}_{\mathrm{in}}$) and output gradient magnitudes ($\mathbf{s}_{\mathrm{out}}$) and re-weight the full-precision matrix accordingly:

$$\mathbf{W}' = \mathbf{s}_{\mathrm{out}}^{\alpha_{\mathrm{out}}} \odot \mathbf{W}_{\mathrm{FP}} \odot \mathbf{s}_{\mathrm{in}}^{\alpha_{\mathrm{in}}}. \quad (6)$$

Finally, after running the iterative SVID, the resulting scales are mapped back to the original domain for training: $\mathbf{g}_i = \mathbf{s}_{\mathrm{out}}^{-\alpha_{\mathrm{out}}} \odot \mathbf{g}_i'$ and $\mathbf{h}_i = \mathbf{s}_{\mathrm{in}}^{-\alpha_{\mathrm{in}}} \odot \mathbf{h}_i'$. This strategy dramatically reduces the initial task loss, ensuring a stable start to QAT (see Algorithm 1 for the full algorithm, with analysis in Table 6 and Figure 5).

## 5 EXPERIMENTS

### 5.1 EXPERIMENTAL SETTINGS

**Setup.** We evaluate RaBiT on recent LLMs including Llama2/3 and Gemma3 (Touvron et al., 2023; AI@Meta, 2024). For QAT, we use a 200M-token subset from a combined WikiText-2 and C4 dataset (Jo et al., 2024). We report perplexity (PPL) on their validation sets (context length: 4096) and the average zero-shot accuracy (QA Avg.) on five common sense reasoning benchmarks (*e.g.,* HellaSwag, PIQA) (Sakaguchi et al., 2021; Zellers et al., 2019; Clark et al., 2018; Bisk et al., 2020), with a detailed breakdown provided in Appendix A.4.

Table 2: **Comparison with state-of-the-art 2-3-bit methods on Llama models.** We report perplexity (PPL ↓) and zero-shot QA Average (↑). For the 2-bit results, the best and runner-up are marked in **bold** and underlined, respectively. RaBiT achieves state-of-the-art (SOTA) performance on Llama2-7B and Llama3-8B, while showing highly competitive results on Llama2-13B.

| Methods | Llama-2-7B | | | | Llama-2-13B | | | | Llama-3-8B | | | |
|---|---|---|---|---|---|---|---|---|---|---|---|---|
| | bit | Wiki2↓ | C4↓ | QA Avg↑ | bit | Wiki2↓ | C4↓ | QA Avg↑ | bit | Wiki2↓ | C4↓ | QA Avg↑ |
| Baseline | 16 | 5.12 | 6.63 | 62.26 | 16 | 4.57 | 6.05 | 65.46 | 16 | 5.75 | 8.32 | 68.66 |
| GPTQ | 2.1 | 50.75 | 36.76 | 39.16 | 2.1 | 43.84 | 23.07 | 43.72 | 2 | 1.21e3 | 4.97e2 | 35.59 |
| EfficientQAT | 2.1 | 6.42 | 8.34 | 57.75 | 2.1 | 5.58 | 7.40 | 62.07 | 2.1 | 8.75 | 12.09 | 60.63 |
| AQLM | 2.3 | 6.29 | 8.56 | 58.57 | 2.2 | 5.41 | 7.20 | 61.58 | 2.3 | 7.23 | 10.32 | 64.12 |
| QuIP# | 2 | 6.19 | 8.16 | 58.23 | 2 | 5.35 | 7.20 | 61.96 | 2 | 8.70 | 12.04 | 63.89 |
| QTIP | 2 | 5.86 | 7.73 | 58.97 | 2 | **5.11** | **6.85** | **62.92** | 2 | 7.52 | 10.76 | 63.88 |
| BitStack | 3 | 6.91 | 9.10 | 56.54 | 3 | 5.90 | 7.86 | 61.06 | 3 | 12.38 | 17.51 | 58.41 |
| | 2 | 29.97 | 34.91 | 40.12 | 2 | 67.98 | 72.60 | 39.38 | 2 | 2.75e3 | 1.93e3 | 36.21 |
| DB-LLM | 2 | 7.23 | 9.62 | 55.12 | 2 | 6.19 | 8.38 | 59.41 | 2 | 12.08 | 16.80 | 50.92 |
| MBOK | 3 | 6.13 | 8.13 | 54.63 | 3 | 5.14 | 6.94 | 62.73 | 3 | 7.81 | 11.29 | 61.08 |
| | 2 | 6.99 | 9.38 | 53.63 | 2 | 5.76 | 7.89 | 60.58 | 2 | 10.74 | 14.61 | 54.41 |
| DBF | 2.3 | 5.81 | 7.69 | 59.84 | 2.3 | 5.15 | 6.85 | 62.53 | 2.3 | 7.22 | 10.34 | 64.84 |
| | 2 | 6.10 | 8.05 | 58.42 | 2 | 5.33 | 7.13 | 61.53 | 2 | 7.78 | 10.99 | 62.90 |
| RaBiT(Ours) | 3 | 5.36 | 7.06 | 63.05 | 3 | 4.84 | 6.51 | 64.09 | 3 | 6.58 | 9.54 | 65.61 |
| | 2 | **5.77** | **7.64** | **61.51** | 2 | 5.15 | 6.95 | 62.10 | 2 | **7.34** | **10.52** | **64.13** |

Table 3: **Comparison with state-of-the-art 2-bit methods on Gemma3 models.** We report perplexity (PPL ↓) and zero-shot QA Average (↑). The context length is 4096. RaBiT consistently achieves SOTA or highly competitive performance, demonstrating its robustness across diverse model architectures.

| Methods | Gemma3-1B | | | | Gemma3-4B | | | | Gemma3-12B | | | |
|---|---|---|---|---|---|---|---|---|---|---|---|---|
| | bit | Wiki2↓ | C4↓ | QA Avg↑ | bit | Wiki2↓ | C4↓ | QA Avg↑ | bit | Wiki2↓ | C4↓ | QA Avg↑ |
| Baseline | 16 | 9.80 | 13.69 | 57.82 | 16 | 6.88 | 10.44 | 67.60 | 16 | 5.50 | 9.28 | 73.45 |
| DBF | 2 | 13.28 | 17.57 | 51.98 | 2 | 8.72 | 12.71 | 60.91 | 2 | 6.97 | 10.60 | 68.37 |
| QTIP | 2 | 13.14 | 17.36 | 50.30 | 2 | 8.31 | 12.21 | **63.47** | 2 | **6.65** | 10.25 | **69.69** |
| RaBiT(Ours) | 2 | **11.27** | **15.54** | **53.18** | 2 | **8.09** | **11.91** | 62.21 | 2 | 6.66 | **10.18** | 68.85 |

**Training Details.** We employ a QAT framework with knowledge distillation (KD) (Hinton et al., 2015; Liu et al., 2024), where the full-precision model serves as the teacher. The objective function combines Kullback–Leibler (KL) divergence loss on the output logits with intermediate MSE losses: $\mathcal{L}_{\text{total}} = \mathcal{L}_{\text{kl}} + \gamma \sum_i \mathcal{L}_{\text{inter},i}$, with $\gamma = 10$ for Llama-family, but $\gamma = 0$ for Gemma3 models to avoid instability from their large activation range. All models are trained for 6 epochs with the Muon optimizer (Jordan et al., 2024) and our proposed function-aware initialization. Full hyperparameters are listed in Appendix A.6.

**Baselines.** We benchmark RaBiT against a comprehensive set of state-of-the-art 2- to 3-bit methods. Baselines include (1) standard methods like GPTQ and EfficientQAT (Frantar et al., 2023; Huang et al., 2024a); (2) high-accuracy but hardware-intensive Vector Quantization (VQ) approaches such as AQLM, QuIP#, and QTIP (Egiazarian et al., 2024; Tseng et al., 2024a;b); and (3) hardware-efficient binary/residual methods like BitStack, DB-LLM, MBOK, and DBF (Wang et al., 2024; Chen et al., 2024; Tran & Nguyen, 2025; Boža & Hradiš, 2025)[2], which are the most direct architectural competitors.

## 5.2 MAIN RESULTS

As shown in Tables 2 and 3, RaBiT consistently redefines the state-of-the-art for 2-bit quantization, demonstrating superior performance across all tested models.

---

[2]For DB-LLM and MBOK, we used our re-implementation

**Dominance over Hardware-Efficient Methods.** RaBiT significantly outperforms other matmul-free binary/residual methods. On Llama2-7B, its 5.77 WikiText-2 PPL is a marked improvement over competitors like MBOK (6.99 PPL) and DBF (6.10 PPL). This performance gap widens on larger models and more complex datasets, underscoring the severe performance penalty incurred by the **inter-path adaptation** that these methods fail to address. The catastrophic failure of some methods, such as BitStack on Llama3-8B (2.75e3 PPL), highlights the instability that RaBiT's principled design successfully overcomes.

**Achieving VQ-Level Accuracy with Binary Efficiency.** More impressively, RaBiT closes the gap with and often surpasses hardware-intensive VQ methods, resolving the historical trade-off between accuracy and efficiency. On Llama2-7B, RaBiT's 5.77 PPL edges out the leading VQ method, QTIP (5.86 PPL), setting a new SOTA for 2-bit quantization. This trend holds for reasoning tasks, where RaBiT's 61.51% QA average on Llama2-7B surpasses all other listed methods, including QTIP's 58.97%, demonstrating superior functional preservation. RaBiT's robustness is further evident on Llama3-8B, where it maintains strong performance (7.34 PPL) while VQ methods like QuIP# suffer from severe degradation (8.70 PPL), showcasing the stability of our training framework.

## 5.3 Ablation Studies

### 5.3.1 Component-wise Contribution Analysis

Channel Importance Scaling (**S**) also yields a consistent, albeit smaller, improvement by prioritizing salient weights. The full RaBiT model, integrating all components, achieves the optimal 5.77 PPL, demonstrating that a residually-coupled training mechanism combined with function-aware initialization is essential for state-of-the-art performance. We performed an ablation study to analyze the contributions of RaBiT's core components: Coupled QAT, Iterative SVID (**I**), and I/O Channel Importance-Scaled Preconditioning (**S**), with results in Table 4.

Table 4: **Ablation on RaBiT** (Llama2-7B PPL). The analysis isolates the impact of Iterative Residual SVID (**I**) and I/O Channel Importance-Scaled Preconditioning (**S**).

| Training Method | I | S | Wiki2 ↓ |
|---|---|---|---|
| Standard QAT | | | 6.55 |
| | ✓ | | 6.21 |
| | | ✓ | 6.31 |
| | ✓ | ✓ | 6.18 |
| **Coupled QAT** **(RaBiT)** | | | 5.83 |
| | ✓ | | 5.78 |
| | | ✓ | 5.81 |
| | ✓ | ✓ | **5.77** |

The analysis clearly shows that **Coupled QAT** is the most critical performance factor. Simply switching from Standard QAT (6.55 PPL) to Coupled QAT drops the perplexity to 5.83, confirming that resolving inter-path adaptation yields the largest gain.

Our initialization methods (**I** and **S**) provide further essential improvements. While they offer a significant boost to the baseline Standard QAT, their role within the powerful Coupled QAT framework is to provide the final, crucial fine-tuning needed to reach the optimal 5.77 PPL. This synergy between a robust training method and a function-aware initialization is key to RaBiT's state-of-the-art performance.

### 5.3.2 Analysis of Coupled Training Dynamics

To empirically validate that coupled training resolves co-adaptation, we conducted a controlled experiment comparing RaBiT to four variants: (1) **Standard QAT** (independent latent weights), (2) **MBOK** (frozen primary binary core, mimicking the path-freezing heuristic of Tran & Nguyen (2025) within our training loop), (3) **Scale-only** (frozen binary cores), and (4) **Scale-frozen** (RaBiT with frozen scales). All variants shared the same initialization and hyperparameters; for a fair comparison of training dynamics, MBOK also used the same optimizer as our model.

Figure 2 reveals the resulting training dynamics. As theorized, **RaBiT** successfully maintains a stable negative inter-path correlation, enforcing the error-correction hierarchy (Figure 2a). In contrast, **Standard QAT** develops a strong positive correlation, confirming that a shared global gradient induces harmful redundancy. The constrained variants (**MBOK**, **Scale-**

**frozen**) fail to establish a strong anti-correlation, limiting their optimization potential. This structural advantage directly translates to model functionality, as shown by the training loss curves (Figure 2b). RaBiT achieves the lowest and most stable loss, while the co-adaptation in Standard QAT and the incomplete optimization of the other variants lead to significantly higher loss. This analysis confirms that RaBiT's ability to jointly optimize all parameters while algorithmically preventing co-adaptation is the key to its superior performance.

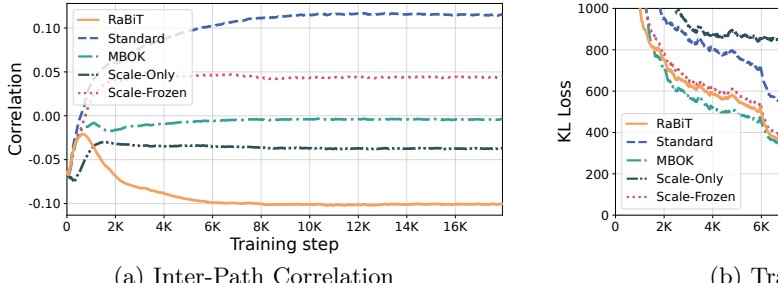

(a) Inter-Path Correlation        (b) Training Loss

Figure 2: **Visualization of Coupled Training Dynamics. (a) Inter-Path Correlation:** RaBiT enforces a negative inter-path correlation, indicating effective error-correction, whereas Standard QAT leads to positive correlation (co-adaptation). **(b) Training Loss:** This structural advantage directly translates to a lower and more stable training loss for RaBiT, demonstrating its superior optimization path.

## 5.4 Inference Performance

RaBiT not only achieves SOTA accuracy but also delivers exceptional inference speed by leveraging its parallelizable matmul-free binary architecture. As shown in Table 5, RaBiT achieves up to a **4.49× speed-up** in end-to-end decoding throughput for a 256-token generation over the FP16 baseline on an NVIDIA RTX 4090.

This performance gain stems from two key advantages. First, the 8× reduction in model size (2-bit vs. 16-bit) dramatically lowers memory bandwidth requirements, which is the primary bottleneck in the autoregressive decoding phase. Second, unlike VQ methods, RaBiT avoids hardware-unfriendly overheads like lookup tables or rotations. Its simple architecture of additions and element-wise scaling translates to higher hardware utilization. This is evident in our kernel-level benchmarks, where RaBiT's specialized kernels exhibit consistently lower latency than both the FP16 baseline and QTIP's VQ kernels. By eliminating computational complexity, RaBiT ensures that theoretical memory savings translate directly into real-world speed, delivering a solution that is both accurate and genuinely efficient. Further details on our kernel design and additional performance benchmarks are provided in Appendix A.5.1 and Appendix A.5.2, respectively.

Table 5: **Inference Performance Analysis on NVIDIA RTX 4090.** Kernel latency for key Llama2-7B/13B layers and Llama2-7B decoding throughput for a 256-token generation. **RaBiT shows superior efficiency** at both the kernel and system levels.

| Method | bit | Kernel-Level Latency ($\mu s$) ↓ | | | | End-to-End Decoding |
|---|---|---|---|---|---|---|
| | | 4096×4096 (q_proj, 7B) | 11008×4096 (gate_proj, 7B) | 5120×5120 (q_proj, 13B) | 13824×5120 (gate_proj, 13B) | Throughput (tok/s) ↑ |
| FP16 | 16 | 17.15 (1.00×) | 70.37 (1.00×) | 17.85 (1.00×) | 122.90 (1.00×) | 64.96 (1.00×) |
| DBF | 2.3 | 12.66 (1.35×) | 28.43 (2.48×) | 14.72 (1.21×) | 31.87 (3.86×) | 157.66 (2.43×) |
| | 2 | 11.47 (1.50×) | 20.90 (3.37×) | 14.08 (1.27×) | 29.58 (4.15×) | 175.21 (2.70×) |
| QTIP | 3 | 24.04 (0.71×) | 37.08 (1.90×) | 36.22 (0.49×) | 49.97 (2.46×) | 153.59 (2.36×) |
| | 2 | 23.40 (0.73×) | 42.40 (1.66×) | 37.46 (0.48×) | 59.22 (2.08×) | 171.74 (2.64×) |
| **RaBiT (Ours)** | 3 | **8.15 (2.10×)** | **17.13 (4.11×)** | **9.90 (1.80×)** | **22.36 (5.50×)** | **191.63 (2.95×)** |
| | 2 | **7.72 (2.22×)** | **15.71 (4.48×)** | **8.33 (2.14×)** | **17.50 (7.02×)** | **291.88 (4.49×)** |

## 6 Conclusion

This paper resolves the critical trade-off between accuracy and hardware efficiency in 2-bit LLM quantization by introducing **RaBiT**. We first identify and analyze **inter-path adaptation** as a fundamental bottleneck that compromises the error-compensation structure in residual binarization. RaBiT's core mechanism, on-the-fly **residual coupling**, algorithmically prevents this breakdown during training, ensuring the model's full expressive power is utilized. To tame the notoriously unstable dynamics of extreme quantization, we further introduce a robust **function-aware initialization** strategy that ensures stable convergence. Our comprehensive experiments demonstrate that RaBiT achieves new **state-of-the-art performance** at 2-bit precision, surpassing not only existing binary methods but also complex and hardware-intensive vector quantization approaches. By establishing this new frontier, RaBiT paves the way for the efficient deployment of high-performance low-bit LLMs and provides a scalable foundation for future research.

## Reproducibility Statement

We are committed to ensuring the reproducibility of our research. To this end, we provide detailed descriptions of our methodology, experimental setup, and hyperparameters throughout the paper and its appendices.

**Algorithm and Implementation Details.** While full source code availability is subject to our organization's review protocols, we have made every effort to ensure reproducibility by providing detailed algorithmic descriptions. The core logic of the **RaBiT** QAT framework is thoroughly explained in Section 4 and presented as step-by-step pseudocode in Algorithm 2. Similarly, the design and optimization principles of our high-performance CUDA kernel are described in Appendix A.5.1, providing a clear blueprint for implementation.

**Data.** Our training process utilizes a 200M-token subset of the publicly available WikiText-2 and C4 datasets, generated using the same data processing approach as described in Jo et al. (2024)[3]. All evaluations are performed on standard public benchmarks (WikiText-2, C4, HellaSwag, PIQA, ARC-e, ARC-c, and WinoGrande), as detailed in Section 5.1.

**Hyperparameters and Infrastructure.** A comprehensive list of all hyperparameters used for training each model, including model-specific learning rates, optimizer settings, and the initialization parameters $(\alpha_{\text{in}}, \alpha_{\text{out}}, T_{\max})$, is provided in Appendix A.6. All experiments were conducted on a single node equipped with four NVIDIA H100 GPUs. The reported results are from a single training run for each model, which is a standard practice for LLM fine-tuning at this scale due to the high computational cost.

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

# A  APPENDIX

## A.1  MATHEMATICAL ANALYSIS OF TRAINING DYNAMICS

This section provides a mathematical analysis of the training dynamics for residual binary architectures. We demonstrate why Standard QAT is prone to **inter-path adaptation**, where paths become redundant. In contrast, we show how RaBiT's coupled training mechanism structurally enforces an **error-correcting hierarchy** and is superior to other heuristic solutions.

**Proposition 1** (Inter-Path Adaptation in Standard QAT)**.** *In a Standard QAT scheme where two paths $(\hat{\mathbf{W}}_1, \hat{\mathbf{W}}_2)$ are updated from their respective latent weights $(\mathbf{W}_1, \mathbf{W}_2)$ using a shared global gradient $\mathbf{G} = \nabla_{\hat{\mathbf{W}}_1 + \hat{\mathbf{W}}_2} \mathcal{L}$, the paths have a persistent tendency to become positively correlated, leading to redundancy.*

*Proof.* Let the latent weights be $\mathbf{W}_1$ and $\mathbf{W}_2$. After a single update step with learning rate $\eta$ and shared gradient $\mathbf{g}$, the new weights are $\mathbf{W}_1'$ and $\mathbf{W}_2'$:

$$\mathbf{W}_1' := \mathbf{W}_1 - \eta \mathbf{G} \quad \text{and} \quad \mathbf{W}_2' := \mathbf{W}_2 - \eta \mathbf{G}$$

The change in the Frobenius inner product between the weights, which reflects their correlation, is:

$$\Delta_{\langle \cdot, \cdot \rangle} := \langle \mathbf{W}_1', \mathbf{W}_2' \rangle_F - \langle \mathbf{W}_1, \mathbf{W}_2 \rangle_F$$

Expanding this gives:

$$\Delta_{\langle \cdot, \cdot \rangle} = -\eta \left( \langle \mathbf{W}_1, \mathbf{G} \rangle_F + \langle \mathbf{W}_2, \mathbf{G} \rangle_F \right) + \eta^2 \|\mathbf{G}\|_F^2$$

While the linear terms depend on the alignment between the current weights and the gradient, the quadratic term $\eta^2 \|\mathbf{G}\|_F^2$ is **always non-negative**. This term acts as a systematic force, constantly pushing the two paths in the same direction defined by the global gradient $\mathbf{G}$. This dynamic, the underlying mechanism of **inter-path adaptation**, compels both paths to learn redundant, dominant features in order to minimize the global loss. This ultimately leads to a break down of the intended residual hierarchy and compromises the model's expressive capacity. $\qquad \square$

**Proposition 2** (Structurally Enforced Error Correction in RaBiT)**.** *RaBiT's coupled training mechanism resolves the redundancy drift by fundamentally changing the optimization objective. Instead of independent updates, RaBiT's on-the-fly derivation structurally forces the second path $(\hat{\mathbf{W}}_2)$ to align with the true residual of the first path $(\mathbf{R}_1 = \mathbf{W}_{\text{FP}} - \hat{\mathbf{W}}_1)$, thereby enforcing an error-correcting relationship.*

*Analysis.* The optimization objectives of the two paths are implicitly different in RaBiT versus the naïve approach.

- **Standard QAT Objective**: Both paths are driven by the same structurally-agnostic global gradient $\mathbf{G}$. Their implicit goal is to align with $\mathbf{G}$ to reduce the global loss. Since both $\hat{\mathbf{W}}_1$ and $\hat{\mathbf{W}}_2$ are incentivized to align with the same vector $\mathbf{G}$, they inevitably learn to align with each other, leading to redundancy as shown in Proposition 1.

$$\hat{\mathbf{W}}_1 \propto \mathbf{G} \quad \text{and} \quad \hat{\mathbf{W}}_2 \propto \mathbf{G} \implies \langle \hat{\mathbf{W}}_1, \hat{\mathbf{W}}_2 \rangle_F > 0$$

- **RaBiT's Enforced Objective**: RaBiT maintains a single shared blueprint, $\mathbf{W}_{\text{FP}}$. The on-the-fly derivation process, $\mathbf{R}_1 := \mathbf{W}_{\text{FP}} - \hat{\mathbf{W}}_1$ followed by the binarization of $\mathbf{R}_1$ to create $\hat{\mathbf{W}}_2$, explicitly defines the optimization target for the second path. The goal for $\hat{\mathbf{W}}_2$ is no longer to align with the global gradient $\mathbf{g}$, but to be the best possible low-rank approximation of the current residual $\mathbf{R}_2$.

$$\text{Objective for } \hat{\mathbf{W}}_2 : \quad \min \|\mathbf{R}_1 - \hat{\mathbf{W}}_2\|_F^2 \implies \hat{\mathbf{W}}_2 \approx \mathbf{R}_1$$

This structural constraint forces a high **Residual Alignment**. In the context of extreme low-bit quantization, the first approximation $\hat{\mathbf{W}}_1$ often "overshoots" the target $\mathbf{W}_{\text{FP}}$ in certain directions. To correct this, the residual $\mathbf{R}_1 = \mathbf{W}_{\text{FP}} - \hat{\mathbf{W}}_1$ will point in the

opposite direction of the overshoot. By aligning with $\mathbf{R}_1$, $\hat{\mathbf{W}}_2$ naturally becomes **anti-correlated** with $\hat{\mathbf{W}}_1$, implementing an efficient **active cancellation** mechanism rather than degenerating into redundancy.

$\square$

**Proposition 3** (Superior Optimization Dynamics of Coupled vs. Iterative Training). *Iterative training (e.g., freezing one path while training the other) avoids adaptation but at the cost of optimization efficiency. In contrast, RaBiT resolves adaptation while permitting full parameter co-adaptation, resulting in a superior optimization trajectory.*

*Proof.* Following Proposition 1, the problem of Standard QAT is the simultaneous update of both paths in the same direction. An alternative solution is to update them iteratively, which prevents this simultaneous push and thus avoids adaptation. However, this introduces a new problem of inefficiency.

The optimal direction to reduce the loss $\mathcal{L}$ is the steepest descent direction in the joint parameter space of $(\mathbf{W}_1, \mathbf{W}_2)$, which is $\mathbf{d}^* = (-\mathbf{G}, -\mathbf{G})$. When training iteratively, one path is frozen, so the update is restricted to an axis-aligned direction, e.g., $\mathbf{d}_{\text{iter}} = (\mathbf{0}, -\mathbf{G})$. The cosine similarity between the iterative update and the optimal update direction is:

$$\cos(\theta) = \frac{\langle \mathbf{d}_{\text{iter}}, \mathbf{d}^* \rangle_F}{\|\mathbf{d}_{\text{iter}}\|_F \|\mathbf{d}^*\|_F} = \frac{\langle (\mathbf{0}, -\mathbf{G}), (-\mathbf{G}, -\mathbf{G}) \rangle_F}{\|(\mathbf{0}, -\mathbf{G})\|_F \|(-\mathbf{G}, -\mathbf{G})\|_F}$$

$$= \frac{\|\mathbf{G}\|_F^2}{\|\mathbf{G}\|_F \cdot \sqrt{\|\mathbf{G}\|_F^2 + \|\mathbf{G}\|_F^2}} = \frac{1}{\sqrt{2}}$$

This fixed 45° misalignment forces the optimization to follow an inefficient zig-zag trajectory. While it solves adaptation, it sacrifices optimization efficiency.

RaBiT, through its coupled derivation described in Proposition 2, resolves this trade-off. By updating a single shared weight $\mathbf{W}_{\text{FP}}$ with the full gradient $\mathbf{G}$, it allows both paths to co-adapt simultaneously in a coordinated manner that is not restricted to an inefficient path. Thus, RaBiT resolves adaptation without compromising optimization efficiency, leading to superior dynamics. $\square$

**Corollary 1 (to Proposition 2). Negative Correlation Induction.** *RaBiT's coupled training mechanism, by forcing the second path ($\hat{\mathbf{W}}_2$) to approximate the residual of the first path ($\mathbf{R}_1$), inherently promotes a negative correlation between their respective outputs ($\mathbf{y}_1, \mathbf{y}_2$).*

*Analysis.* From Proposition 2, we established that RaBiT trains the second path to approximate the residual of the first:

$$\hat{\mathbf{W}}_2 \approx \mathbf{R}_1 = \mathbf{W}_{\text{FP}} - \hat{\mathbf{W}}_1$$

Let's consider the outputs for a given input $\mathbf{x}$. The outputs of the full-precision teacher, the first path, and the second path are $\mathbf{y}_t = \mathbf{W}_{\text{FP}}\mathbf{x}$, $\mathbf{y}_1 = \hat{\mathbf{W}}_1\mathbf{x}$, and $\mathbf{y}_2 = \hat{\mathbf{W}}_2\mathbf{x}$, respectively. Based on the weight approximation, the output of the second path is:

$$\mathbf{y}_2 \approx \mathbf{R}_1\mathbf{x} = (\mathbf{W}_{\text{FP}} - \hat{\mathbf{W}}_1)\mathbf{x} = \mathbf{y}_t - \mathbf{y}_1$$

Now, we can analyze the covariance between the outputs $\mathbf{y}_1$ and $\mathbf{y}_2$. Assuming the outputs are centered for simplicity, the covariance is proportional to the expected value of their dot product, $\mathbb{E}[\mathbf{y}_1^\top \mathbf{y}_2]$.

$$\mathbb{E}[\mathbf{y}_1^\top \mathbf{y}_2] \approx \mathbb{E}[\mathbf{y}_1^\top (\mathbf{y}_t - \mathbf{y}_1)] = \mathbb{E}[\mathbf{y}_1^\top \mathbf{y}_t] - \mathbb{E}[\mathbf{y}_1^\top \mathbf{y}_1] = \mathbb{E}[\mathbf{y}_1^\top \mathbf{y}_t] - \mathbb{E}[\|\mathbf{y}_1\|^2]$$

Let's analyze the two terms:

1. $\mathbb{E}[\mathbf{y}_1^\top \mathbf{y}_t]$: The first path $\hat{\mathbf{W}}_1$ is the primary, albeit coarse, approximation of $\mathbf{W}_{\text{FP}}$. Its purpose is to capture the main features of the teacher. Therefore, their outputs $\mathbf{y}_1$ and $\mathbf{y}_t$ are expected to be strongly and **positively correlated**, making this term a large positive value.

2. $\mathbb{E}[\|\mathbf{y}_1\|^2]$: This is the expected squared norm of the first path's output. Binarization is an aggressive quantization that often leads to an "overshoot" in magnitude. A single binary path must represent a wide range of continuous values, so its effective scaling factor often results in an output magnitude $\|\mathbf{y}_1\|$ that exceeds the projection of $\mathbf{y}_1$ onto $\mathbf{y}_t$. Consequently, it is generally the case that $\|\mathbf{y}_1\|^2 > \mathbf{y}_1^\top \mathbf{y}_t$, making $\mathbb{E}[\|\mathbf{y}_1\|^2]$ a larger positive term than $\mathbb{E}[\mathbf{y}_1^\top \mathbf{y}_t]$.

Combining these points, the covariance is approximately the difference between a positive term and a larger positive term:

$$\text{Cov}(\mathbf{y}_1, \mathbf{y}_2) \approx \underbrace{\mathbb{E}[\mathbf{y}_1^\top \mathbf{y}_t]}_{\text{Positive Alignment}} - \underbrace{\mathbb{E}[\|\mathbf{y}_1\|^2]}_{\text{Larger Magnitude Term}} < 0$$

Thus, RaBiT's mechanism of forcing the second path to correct the error of the first path structurally drives the covariance, and therefore the correlation $\text{Corr}(\mathbf{y}_1, \mathbf{y}_2)$, to be negative. $\qquad\square$

### A.2 Extended Analysis: Inter-Path Adaptation under KL Divergence

While Section 3 and Corollary 1 motivate RaBiT using the Mean Squared Error (MSE) decomposition, modern LLM training often relies on the Kullback-Leibler (KL) divergence loss. In this section, we rigorously demonstrate that the principle of "inter-path adaptation" and RaBiT's solution remain valid under the KL divergence objective. We leverage a local quadratic approximation and the findings from Kim et al. (2021) regarding the decomposition of KL loss.

**Proposition 4** (Optimality of Residual Coupling under KL Divergence). *RaBiT's residual coupling mechanism structurally eliminates the optimization bias inherent in the KL divergence loss by enforcing a negative Hessian-weighted path correlation, a property that Standard QAT fails to satisfy.*

*Analysis.* We analyze the optimization dynamics at a specific linear layer $\ell$ where RaBiT is applied. Let $\mathbf{y}_t^\ell = \mathbf{W}_{\text{FP}}\mathbf{x}^\ell$ be the output feature of the teacher model, and $\mathbf{y}_s^\ell$ be that of the student. Since the operations within the layer are linear, the student's output is the sum of its binary paths: $\mathbf{y}_s^\ell = \mathbf{h}_1 + \mathbf{h}_2$.

**Local Quadratic Approximation.** The global KL divergence loss $\mathcal{L}_{KL}$ can be approximated locally around the teacher's output $\mathbf{y}_t^\ell$ using a second-order Taylor expansion:

$$\mathcal{L}_{KL}(\mathbf{y}_s^\ell) \approx \mathcal{L}_{KL}(\mathbf{y}_t^\ell) + \nabla\mathcal{L}(\mathbf{y}_t^\ell)^\top \Delta\mathbf{y} + \frac{1}{2}\Delta\mathbf{y}^\top \mathbf{H}^\ell \Delta\mathbf{y} \tag{7}$$

Assuming the teacher is optimal locally ($\nabla\mathcal{L} \approx 0$), the optimization objective reduces to minimizing a **Hessian-weighted MSE**:

$$\mathcal{J}_{local} \approx \frac{1}{2}\|\mathbf{y}_s^\ell - \mathbf{y}_t^\ell\|_{\mathbf{H}^\ell}^2 = \frac{1}{2}(\mathbf{h}_1 + \mathbf{h}_2 - \mathbf{y}_t^\ell)^\top \mathbf{H}^\ell (\mathbf{h}_1 + \mathbf{h}_2 - \mathbf{y}_t^\ell) \tag{8}$$

where $\mathbf{H}^\ell$ is the Hessian matrix representing the local curvature.

**Decomposition with Hessian-weighted Path Correlation.** Similar to Eq. (3) in the main text, we decompose this local objective. By treating $(\mathbf{h}_1 - \mathbf{y}_t^\ell)$ as the residual error of the first path, we expand the quadratic term:

$$\mathcal{J}_{local} \propto \underbrace{\|\mathbf{h}_1 - \mathbf{y}_t^\ell\|_{\mathbf{H}^\ell}^2}_{\text{Base Error}} + \underbrace{\|\mathbf{h}_2\|_{\mathbf{H}^\ell}^2}_{\text{Path 2 Amp.}} + \underbrace{2(\mathbf{h}_1 - \mathbf{y}_t^\ell)^\top \mathbf{H}^\ell \mathbf{h}_2}_{\text{Interaction Term}} \tag{9}$$

We define the *Hessian-weighted Path Correlation*, denoted as $\text{PathCorr}_{\mathbf{H}}$, based on the generalized cosine similarity in the inner product space defined by $\mathbf{H}^\ell$:

$$\text{Interaction Term} = 2 \cdot \|\mathbf{h}_1 - \mathbf{y}_t^\ell\|_{\mathbf{H}^\ell}\|\mathbf{h}_2\|_{\mathbf{H}^\ell} \cdot \mathbf{PathCorr_H}(\mathbf{h}_1 - \mathbf{y}_t^\ell, \mathbf{h}_2) \tag{10}$$

This formulation reveals that minimizing the local loss requires maximizing the negative magnitude of $\text{PathCorr}_{\mathbf{H}}$.

**Bridging Interaction to Bias Cancellation.** To understand the implication of this interaction term in the context of KL divergence, we refer to Kim et al. (2021), which proved that minimizing $\mathcal{L}_{KL}$ is equivalent to minimizing MSE plus a negative **Bias Term ($\delta$)**. This bias term acts as a destabilizing force that pushes the student's logit sum to diverge from the teacher's. Crucially, effectively maximizing the negative interaction term (i.e., enforcing error correction) is the key to neutralizing this bias.

- **Failure of Standard QAT:** In Standard QAT, $\mathbf{h}_1$ and $\mathbf{h}_2$ are updated independently using the same gradient signal. This leads to $\mathbf{h}_2$ aligning with $\mathbf{y}_t^\ell$ rather than the residual $(\mathbf{y}_t^\ell - \mathbf{h}_1)$. Consequently, PathCorr$_\mathbf{H}$ becomes positive (redundancy) or near zero. This failure to exploit the interaction bonus leaves the destabilizing Bias Term ($\delta$) unchecked, causing the optimization drift described in Kim et al. (2021).

- **Success of RaBiT:** RaBiT enforces $\hat{\mathbf{W}}_2 = \text{sign}(\mathbf{W}_{\text{FP}} - \hat{\mathbf{W}}_1)$, structurally guaranteeing:
$$\mathbf{h}_2 \approx \mathbf{y}_t^\ell - \mathbf{h}_1 = -(\mathbf{h}_1 - \mathbf{y}_t^\ell) \tag{11}$$
This forces the second path vector to be anti-parallel to the first path's error vector in the feature space. As a result:

  1. **Maximized Interaction:** PathCorr$_\mathbf{H}$ approaches its theoretical minimum of $-1$ (perfect anti-correlation).
  2. **Bias Cancellation:** By strictly adhering to the residual $\mathbf{h}_2 \approx \mathbf{y}_t^\ell - \mathbf{h}_1$, the total student output $\mathbf{y}_s^\ell$ approximates $\mathbf{y}_t^\ell$ without the scale divergence issues. The Bias Term ($\delta$) is effectively cancelled out locally $((\sum \mathbf{y}_s^\ell - \sum \mathbf{y}_t^\ell)^2 \approx 0)$.

Therefore, RaBiT's residual coupling is the optimal strategy for minimizing $\mathcal{L}_{KL}$ as it structurally enforces the necessary negative correlation that Standard QAT fails to learn. □

### A.3 INITIALIZATION ANALYSIS: FUNCTIONALITY VS. APPROXIMATION

Table 6: **Initialization Analysis on Llama2-7B.** Trade-off between weight reconstruction error (Avg. MAE/MSE) and model functionality (Initial KL Divergence Loss), for the first `q_proj` layer. I/O Channel Importance Scaling dramatically reduces KL Divergence Loss despite increasing MSE.

| Initialization Method | Avg. MAE ↓ | Avg. MSE ↓ | KL Loss ↓ |
|---|---|---|---|
| Greedy SVID | 0.359 | 0.150 | 17,152 |
| Iterative Residual SVID | 0.370 | 0.122 | 13,760 |
| + I/O Ch. Importance Scaling | **0.632** | **0.302** | **2,672** |

Stable initialization is paramount in the low-bit regime, as the initial quantization error spike can destabilize QAT. On the Llama2-7b model, we evaluate our proposed techniques—Iterative Residual SVID and I/O Channel Importance Scaling (Section 4.3)—by measuring both the weight reconstruction error (Avg. MAE, MSE) and the initial task loss (Knowledge Distillation (KD) loss) before the first training step.

Table 6 details the results averaged across attention projection layers and reveals a crucial insight. As our baseline, Greedy SVID is a non-iterative decomposition that finalizes each path sequentially without the co-adaptation enabled by our iterative approach. First, regarding **Iterative Refinement**, moving from Greedy SVID to Iterative Residual SVID consistently improves weight reconstruction (*e.g.,* Avg. MSE drops $0.150 \rightarrow 0.122$) and substantially reduces the initial KL divergence loss ($17{,}152 \rightarrow 13{,}760$), confirming mitigation of scheduling bias. Second, adding **I/O Channel Importance Scaling** to the iterative process yields a striking result: while reconstruction error increases significantly (Avg. MSE $0.122 \rightarrow 0.302$), the KL divergence loss *plummets* dramatically ($13{,}760 \rightarrow 2{,}672$, an 81% reduction).

This confirms that extreme quantization should prioritize preserving *functionality* over merely approximating *weights*. I/O Channel Importance Scaling allocates the limited 2-bit capacity

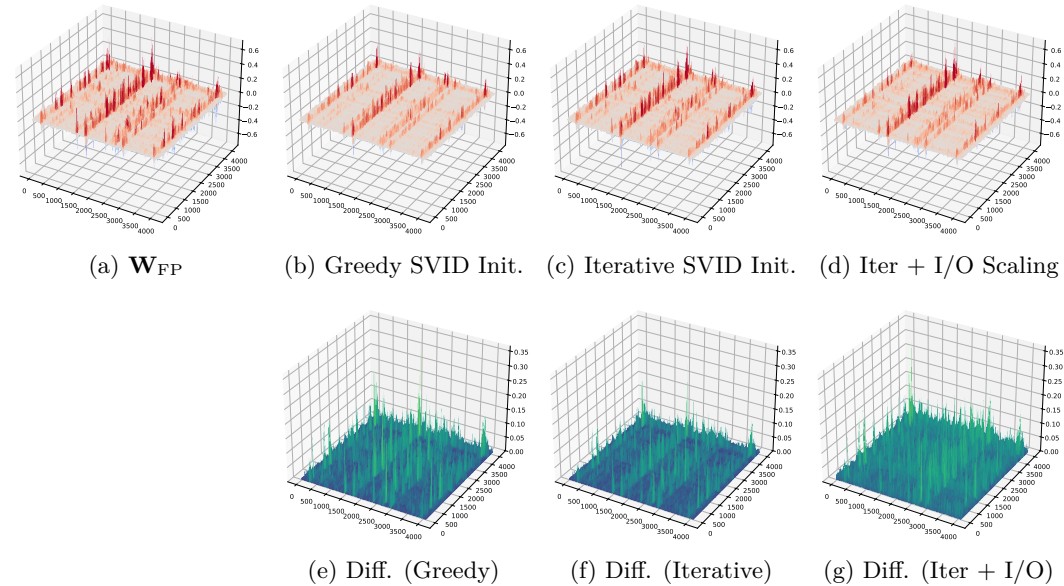

(a) $\mathbf{W}_{\mathrm{FP}}$    (b) Greedy SVID Init.    (c) Iterative SVID Init.    (d) Iter + I/O Scaling

(e) Diff. (Greedy)    (f) Diff. (Iterative)    (g) Diff. (Iter + I/O)

Figure 3: **Visual analysis of weight initialization for the first layer's `q_proj` matrix of the Llama2-7B model.** The top row displays the original full-precision weight ($\mathbf{W}_{\mathrm{FP}}$) alongside its initial approximations from three methods: (b) Greedy SVID, (c) Iterative SVID, and (d) Iterative SVID with I/O Channel Importance Scaling. The bottom row shows the corresponding difference matrices ($\mathbf{W}_{\mathrm{FP}} - \hat{\mathbf{W}}_{\mathrm{init}}$), illustrating the initial error structure. Our function-aware initialization produces a qualitatively different structure compared to the others, which is reflected in its distinct error pattern.

to critical channels based on activation and gradient statistics (Section 4.3), sacrificing the reconstruction of less important weights. This trade-off is visually stark in Figure 3. While Iterative SVID produces a lower-error approximation than Greedy SVID (comparing Figure 3f to Figure 3e), the function-aware I/O Scaling method yields a visibly larger reconstruction error (Figure 3g). Despite this higher weight-level discrepancy, its focus on functional saliency provides a far superior starting point for QAT, as evidenced by the dramatic reduction in initial task loss.

## A.4 Extended Results

**Detailed Zero-Shot Reasoning Accuracy.**    Table 7 and Table 8 provide a detailed breakdown of the zero-shot reasoning accuracy across five common benchmarks, complementing the average scores reported in the main text.

Table 7: **Detailed Zero-Shot Reasoning Accuracy on Llama Models** (%). Comparison of FP16 against leading 2-bit methods on five common benchmarks.

| Models | Method | WinoGrande↑ | HellaSwag↑ | ARC-e↑ | ARC-c↑ | PIQA↑ | Average↑ |
|---|---|---|---|---|---|---|---|
| Llama2-7B | FullPrecision | 67.80 | 56.71 | 69.28 | 39.93 | 78.29 | 62.40 |
| | QTIP | 64.64 | 53.09 | 65.57 | 35.67 | **75.90** | 58.97 |
| | DBF | 63.61 | 52.44 | 64.73 | 35.58 | 75.84 | 58.44 |
| | RaBiT (Ours) | **67.80** | **53.52** | **72.43** | **37.88** | **75.90** | **61.51** |
| Llama2-13B | FullPrecision | 69.93 | 59.64 | 73.19 | 45.73 | 78.67 | 65.43 |
| | QTIP | **67.56** | **57.4** | **70.8** | **41.46** | 77.37 | **62.92** |
| | DBF | 67.09 | 56.6 | 69.02 | 38.74 | **78.18** | 61.93 |
| | RaBiT (Ours) | **67.56** | 56.71 | 69.06 | 39.76 | 77.42 | 62.10 |
| Llama3-8B | FullPrecision | 72.93 | 60.08 | 80.30 | 50.17 | 79.76 | 67.80 |
| | QTIP | **70.24** | **55.53** | 75.29 | 41.64 | 76.71 | 63.88 |
| | DBF | 68.90 | 54.49 | 74.62 | 39.76 | 76.44 | 62.84 |
| | RaBiT (Ours) | 69.37 | 55.13 | **75.37** | **42.83** | **77.96** | **64.13** |

Table 8: **Detailed Zero-Shot Reasoning Accuracy on Gemma Models** (%). Comparison of FP16 against leading 2-bit methods on five common benchmarks.

| Models | Method | WinoGrande↑ | HellaSwag↑ | ARC-e↑ | ARC-c↑ | PIQA↑ | Average↑ |
|---|---|---|---|---|---|---|---|
| Gemma3-1B | FullPrecision | 59.59 | 47.30 | 72.22 | 35.32 | 74.65 | 57.82 |
| | QTIP | 54.62 | 38.24 | 63.93 | 25.85 | 68.88 | 50.30 |
| | DBF | **58.01** | 40.37 | 62.92 | 28.41 | 70.18 | 51.98 |
| | RaBiT (Ours) | 56.59 | **42.94** | **64.52** | **29.44** | **72.42** | **53.18** |
| Gemma3-4B | FullPrecision | 69.22 | 56.77 | 81.52 | 51.45 | 79.05 | 67.60 |
| | QTIP | **66.85** | 52.25 | **77.53** | **44.62** | 76.12 | **63.47** |
| | DBF | 63.69 | 50.15 | 74.74 | 40.87 | 75.08 | 60.91 |
| | RaBiT (Ours) | 65.19 | **52.57** | 75.04 | 41.38 | **76.88** | 62.21 |
| Gemma3-12B | FullPrecision | 75.45 | 61.98 | 87.08 | 61.60 | 81.12 | 73.45 |
| | QTIP | **72.69** | 57.99 | **84.09** | **54.95** | 78.73 | **69.69** |
| | DBF | 72.14 | 57.20 | 82.49 | 52.05 | 77.97 | 68.37 |
| | RaBiT (Ours) | 72.30 | **58.45** | 82.41 | 52.13 | **78.95** | 68.85 |

## A.5 Inference Performance Analysis

### A.5.1 Kernel Design

Our CUDA kernels implement binary GEMV operations tailored to the memory-bound regime typical of the decoding phase in LLM inference. The design centers on bit-packing to reduce global memory traffic, with a latency-tolerant and matmul-free compute pipeline that leverages register-level staging.

**Weight Packing.** To reduce memory traffic, each group of 32 columns is mapped to a `uint32_t`, with $+1 \mapsto 0$ and $-1 \mapsto 1$. We then group the 32-bit words into `uint2` or `PackedBits3` (3 `uint32_t` weights with padding), for 2-bit (2 binary weights) and 3-bit (3 binary weights) models, respectively. Rows are interleaved into warp-sized groups, ensuring that a warp issues full coalesced memory transactions when loading weights. Our efficient packing reduces the raw footprint of weights by a factor of $32\times$, compared to full-precision weights.

**Compute Pipeline.** Each warp is assigned a set of output rows to avoid inter-warp synchronization. Input activations ($\mathbf{x}$) and column scales ($\mathbf{g}$) are read as vectorized `uint4` chunks. Binary signs are applied via lane-local bit shifts and XOR masks, instead of matrix multiplication. The kernel uses simple yet effective pipelining: while one tile of data is consumed, the subsequent tile is prefetched into registers. Accumulation proceeds using `half2` fused multiply-add intrinsics (`__hfma2`), which increase arithmetic throughput without resorting to shared memory. Finally, reductions across threads in a warp are performed with shuffle operations, and output scale factors ($\mathbf{h}$) are applied in `fp16` precision. Notably, our architecture enables per-path parallelizable computation - instead of an n-bit weight, we parallelize with n 1-bit operations.

Efficient weight packing and pipelining reduce global memory access and raise the utilization of execution units on the GPU. The kernel therefore shifts the limiting factor from raw memory bandwidth toward register throughput, yielding measurable efficiency gains during the decoding stage of LLM inference, showing remarkable performance.

### A.5.2 More Comparisons

To provide a more detailed analysis of the end-to-end inference speed, we benchmarked RaBiT against several key baselines: the full-precision (FP16) model, QTIP as the state-of-the-art Vector Quantization (VQ) method, and DBF, which features a similar stacked binary architecture. For QTIP, we utilized the publicly available CUDA kernels from the official implementation[4]. For DBF, which also uses a stacked binary design but executes its two paths sequentially, we developed an optimized CUDA kernel that runs approximately 21% faster than their public Triton-based implementation to ensure a fair and robust comparison. All evaluations were conducted on an NVIDIA RTX 4090 using the Llama2-7B model.

---

[4] https://github.com/Cornell-RelaxML/qtip

The results, depicted in Figure 4, were benchmarked across a range of generated token lengths (64, 128, 256, 512, and 1024) for a comprehensive analysis. As expected, all 2-bit methods significantly outperform the FP16 baseline due to the $8\times$ reduction in memory bandwidth requirements. More importantly, **RaBiT demonstrates a substantial performance advantage, achieving nearly twice the decoding throughput** of the other 2-bit quantization methods. This speed-up stems directly from the efficiency of our parallel, matmul-free architecture. Unlike DBF, which is bottlenecked by its sequential computation of two binary paths, RaBiT's fully parallel design allows it to maximally leverage the benefits of its efficient binary cores. While the absolute tokens/second rate naturally decreases with longer generation sequences, the relative performance gap between the methods remains consistent, confirming the robustness of RaBiT's architectural advantage.

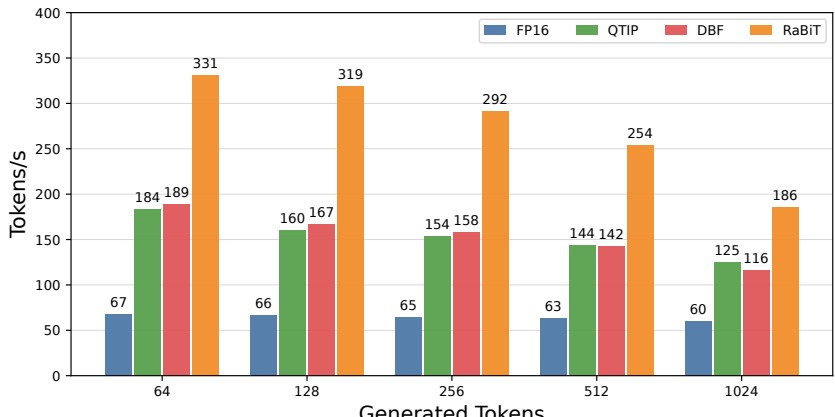

Figure 4: **End-to-end decoding throughput (tokens/second) for Llama2-7B on an NVIDIA RTX 4090 across various generated token lengths.** RaBiT's parallel architecture consistently delivers superior performance over other 2-bit methods.

## A.6 Hyperparameters

### A.6.1 Training Details

We detail the hyperparameters used for our Quantization-Aware Training (QAT) experiments in Table 9. All models were trained for 6 epochs using the Muon optimizer (Jordan et al., 2024) with a cosine learning rate decay schedule. The models were initialized using our proposed function-aware strategy, with a fixed SVID iteration count of $T_{\max} = 20$. Key hyperparameters, such as the learning rate and the I/O Channel Importance Scaling intensities $(\alpha_{\text{in}}, \alpha_{\text{out}})$, were fine-tuned for each specific model to achieve the best performance. All experiments were conducted on a single node equipped with four NVIDIA H100 GPUs.

Table 9: **RaBiT** Training Details

| Bit | Training Setup | Llama2 7B | Llama2 13B | Llama3 8B | Gemma3 1B | Gemma3 4B | Gemma3 12B |
|---|---|---|---|---|---|---|---|
| 2 | Intensities $(\alpha_{\text{in}}, \alpha_{\text{out}})$ | (0.8, 0.65) | (0.95, 0.45) | (0.85, 0.7) | (0.85, 0.7) | (0.95, 0.7) | (0.75, 0.6) |
| | Iteration $(T_{\max})$ | 20 | 20 | 20 | 20 | 20 | 20 |
| | Learning Rate | 12e-6 | 1e-5 | 1e-5 | 1e-5 | 1e-5 | 5e-6 |
| | Epoch | 6 | 6 | 6 | 6 | 6 | 6 |
| | # GPUs | $1 \times 4$ | $1 \times 4$ | $1 \times 4$ | $1 \times 4$ | $1 \times 4$ | $1 \times 4$ |
| | # Training Hours | 39 | 56 | 38 | 8 | 23 | 67 |
| 3 | Intensities $(\alpha_{\text{in}}, \alpha_{\text{out}})$ | (0.8, 0.65) | (0.95, 0.45) | (0.85, 0.7) | - | - | - |
| | Iteration $(T_{\max})$ | 20 | 20 | 20 | - | - | - |
| | Learning Rate | 1e-5 | 1e-5 | 1e-5 | - | - | - |
| | Epoch | 6 | 6 | 6 | - | - | - |
| | # GPUs | $1 \times 4$ | $1 \times 4$ | $1 \times 4$ | - | - | - |
| | # Training Hours | 46 | 88 | 44 | - | - | - |

### A.6.2 Grid Search for I/O Channel Importance Scaling Intensities

To determine the optimal intensity hyperparameters for our I/O Channel Importance Scaling (Section 4.3), we performed a comprehensive grid search. The objective was to identify the values of $\alpha_{\text{in}}$ and $\alpha_{\text{out}}$ that minimized the initial Knowledge Distillation (KD) loss post-initialization. This process utilized a calibration dataset of 128 samples randomly selected from the training data to measure the loss.

The example results of this search on the Llama2-7B model are detailed in Table 10. We observed a clear optimum, with the minimum initial KL divergence loss of 2,672 achieved at the configuration of $\alpha_{\text{in}} = 0.80$ and $\alpha_{\text{out}} = 0.65$. This finding underscores the importance of a balanced preconditioning strategy that considers both input activation statistics and output gradient magnitudes. **We repeated this grid search process for all other models to find their optimal alpha values.**

Table 10: **Grid search results for I/O Channel Importance Scaling Intensities ($\alpha_{\text{in}}$, $\alpha_{\text{out}}$) on Llama2-7B.** The metric is the Initial KL Divergence Loss (Lower is better). The optimal configuration is highlighted in bold.

| $\alpha_{\text{out}}$ | $\alpha_{\text{in}}$ | | | |
|---|---|---|---|---|
| | 0.75 | **0.80** | 0.85 | 0.90 |
| 0.55 | 3,100 | 2,932 | 2,984 | 3,108 |
| 0.60 | 2,932 | 2,938 | 2,971 | 3,143 |
| **0.65** | 3,167 | **2,672** | 2,697 | 3,063 |
| 0.70 | 3,083 | 2,821 | 2,983 | 3,462 |

### A.6.3 SVID Iteration Convergence Analysis

The Iterative Residual SVID initialization (Section 4.3) aims to mitigate the scheduling bias inherent in standard greedy initialization. We analyzed the required number of iterations ($T_{\max}$) for convergence on the Llama2-7B model. We measured the Initial KL divergence loss as the iterations progressed from 1 (equivalent to Greedy SVID) up to 35.

The results, shown in Figure 5, indicate that the initialization quality improves rapidly in the initial phase. The loss stabilizes significantly around 15 iterations, and the optimum is reached at 20 iterations. Beyond this point, further iterations do not provide additional benefits. Based on this analysis, we selected $T_{\max} = 20$ as the default setting for RaBiT initialization, providing an optimal balance between initialization quality and computational cost.

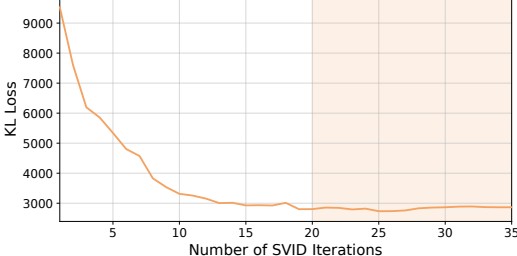

Figure 5: **Convergence analysis of Iterative Residual SVID on Llama2-7B**. The metric is the Initial KL Divergence Loss (Lower is better). Convergence stabilizes around 20 iterations.

### A.6.4 Robustness to Random Seed and Optimizers

To ensure the reliability of RaBiT, we provide additional analyses on training stability across random seeds and optimizer sensitivity (AdamW vs. Muon).

**Multi-Seed Stability.** We conducted experiments on Llama2-7B using 5 different random seeds to verify reproducibility. As shown in Table 11, RaBiT exhibits exceptional stability with a negligible standard deviation in both Perplexity (PPL) and Zero-shot QA accuracy, confirming that our results are robust and not artifacts of a specific seed.

Table 11: **Multi-Seed Stability Analysis (Llama2-7B, 2-bit).** RaBiT demonstrates consistent performance across five random seeds.

| Metric | Seed 123 | Seed 1019 | Seed 1024 | Seed 1112 | Seed 1204 | Avg. (Std.) |
|---|---|---|---|---|---|---|
| Wiki2 PPL ($\downarrow$) | 5.77 | 5.77 | 5.77 | 5.78 | 5.78 | **5.77** (0.005) |
| C4 PPL ($\downarrow$) | 7.63 | 7.62 | 7.63 | 7.63 | 7.64 | **7.63** (0.006) |
| QA Avg. ($\uparrow$) | 62.71 | 62.42 | 62.61 | 62.81 | 62.74 | **62.66** (0.14) |

**Optimizer Sensitivity (AdamW vs. Muon).** While our main experiments utilize the Muon optimizer for memory efficiency, we validated RaBiT with the standard AdamW optimizer to rule out optimizer dependency. On Llama2-7B, RaBiT trained with **AdamW achieved a PPL of 5.78**, which is statistically identical to the **5.77 PPL** obtained with Muon. This confirms that the performance gains stem from the coupled residual architecture, not the choice of optimizer.

A.7 EXTENDED ANALYSIS OF INTER-PATH ADAPTATION

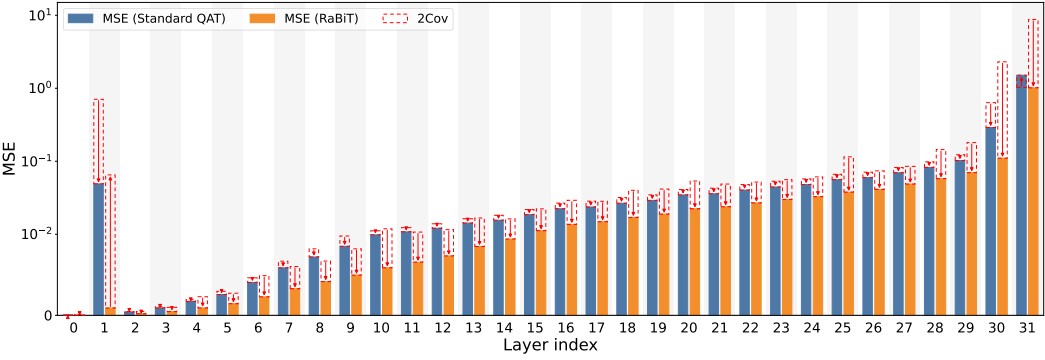

Figure 6: **Layer-wise MSE Decomposition in Llama2-7B's `down_proj` layers.** The bars compare the total Mean Squared Error (MSE) for Standard QAT (blue) and RaBiT (orange). The top of each bar represents the base error term ($C$), while the red-dashed component visualizes twice the covariance ($2 \times \text{Cov}$). RaBiT consistently generates a large negative covariance, which actively reduces the total MSE, demonstrating effective error cancellation. Notably, RaBiT also suppresses the extremely high MSE peak observed in the early layers of the Standard QAT baseline, indicating its robustness against layer sensitivity.

To provide a more granular view of the training dynamics, we conduct a layer-wise analysis of the Mean Squared Error (MSE) decomposition for the Llama2-7B model, visualized in Figure 6. This analysis offers two key insights into RaBiT's structural advantages over Standard QAT.

First, the results empirically confirm our central hypothesis across the network's depth. For most layers, RaBiT consistently generates a substantial negative covariance (the red-dashed component), which acts as a significant loss-reducing bonus, thereby lowering the total MSE. In contrast, Standard QAT fails to establish this effective error-cancellation, exhibiting a much smaller covariance term that provides negligible benefit. This provides strong visual evidence that RaBiT's coupled training successfully enforces the intended error-correction hierarchy, while Standard QAT suffers from the performance degradation of inter-path adaptation.

Second, and more strikingly, the analysis reveals RaBiT's ability to overcome a critical optimization challenge in extreme quantization: layer sensitivity. The Standard QAT baseline exhibits an exceptionally high MSE in the initial layers, a phenomenon consistent with the known sensitivity of early network layers to input distributions and quantization errors, as also observed in other LLMs by (Zhang et al., 2025; 2024). RaBiT, however, dramatically suppresses this MSE peak. This suggests its benefits extend beyond merely enforcing anti-correlation. The fact that RaBiT tames this instability indicates that our method may resolve a more fundamental bottleneck in MSE-based QAT that has historically hindered extreme quantization in conventional architectures. While we designed RaBiT to foster negative correlation, its success in stabilizing these sensitive layers points to a deeper robustness. A full investigation into how residual coupling imparts this stability is a compelling direction for future research.

## A.8 GENERATED SAMPLES

### SAMPLE 1: NEWTON

**Prompt:** Newton discovered the law of universal gravitation. Universal gravitation is

---

Generated Output

---

Newton discovered the law of universal gravitation. Universal gravitation is the force of gravity which draws objects towards the earth and other massive bodies. Gravity is the universal attraction between two objects and is proportional to the product of the masses of the two objects and inversely proportional to the square of the distance between them. Newton explained the force of gravity as being caused by the presence of matter in the universe. He stated that all matter attracted all other matter and that the tendency to move towards a massive body was proportional to the product of the masses of the objects. Newton developed the law of universal gravitation so that the attraction between two bodies could be calculated.

---

### SAMPLE 2: ADAM SMITH

**Prompt:** Adam Smith wrote The Wealth of Nations. The Wealth of Nations is

---

Generated Output

---

Adam Smith wrote The Wealth of Nations. The Wealth of Nations is the first book to outline the modern principles of capitalism and to make the case for it. Smith's work is widely considered to be the first comprehensive system of political economy and one of the most important works of political philosophy. The Wealth of Nations is the foundation of modern capitalism. The Wealth of Nations is a book by Adam Smith, first published in 1776, which describes the nature of economic systems. Adam Smith is regarded as the father of modern economics. The Wealth of Nations is widely considered to be the first comprehensive system of political economy and one of the most important works of political philosophy. It is also one of the most influential books on economics. Adam Smith is regarded as the father of modern economics.

---

### SAMPLE 3: DNA DOUBLE HELIX

**Prompt:** Watson and Crick discovered the DNA double helix. The DNA double helix is

---

Generated Output

---

Watson and Crick discovered the DNA double helix. The DNA double helix is a structure found in the nucleus of every living cell. The double helix is the most fundamental structure in DNA. The structure is made of two single strands of DNA. The double helix was discovered in 1953 by James Watson and Francis Crick. In 1953, James Watson and Francis Crick discovered the double helix structure of DNA. The double helix structure is the most fundamental structure in DNA. The double helix structure is made of two single strands of DNA. The double helix is found in the nucleus of every living cell.

---

## A.9 Algorithms

---

**Algorithm 1** RaBiT Initialization: Calibrated Iterative Residual SVID

---

1: **Require:** Pretrained weight $\mathbf{W}_{\mathrm{FP}}$, Number of paths $k$, Max iterations $T_{\max}$
2: **Require:** Calibration stats $\mathbf{s}_{\mathrm{in}}, \mathbf{s}_{\mathrm{out}}$ and intensities $\alpha_{\mathrm{in}}, \alpha_{\mathrm{out}}$
3: **Output:** Initialized scales $\{(\mathbf{g}_i, \mathbf{h}_i)\}_{i=1}^k$

4: *// Step 1: I/O Channel Importance-Calibrated Preconditioning*
5: Normalize: $\mathbf{s}_{\mathrm{in}} \leftarrow \mathbf{s}_{\mathrm{in}}/\max(\mathbf{s}_{\mathrm{in}}), \quad \mathbf{s}_{\mathrm{out}} \leftarrow \mathbf{s}_{\mathrm{out}}/\max(\mathbf{s}_{\mathrm{out}})$
6: Precondition: $\mathbf{W}' \leftarrow \mathbf{s}_{\mathrm{out}}^{\alpha_{\mathrm{out}}} \odot \mathbf{W}_{\mathrm{FP}} \odot \mathbf{s}_{\mathrm{in}}^{\alpha_{\mathrm{in}}}$

7: *// Step 2: Iterative Residual SVID*
8: Initialize $\hat{\mathbf{W}}_i'^{(0)} \leftarrow \mathbf{0}$ for $i = 1, \dots, k$
9: **for** $t = 1$ to $T_{\max}$ **do**
10:     **for** $i = 1$ to $k$ **do**
11:         *// Calculate target residual (Gauss-Seidel style update)*
12:         $\mathbf{R}_i'^{(t)} \leftarrow \mathbf{W}' - \left(\sum_{j<i} \hat{\mathbf{W}}_j'^{(t)} + \sum_{j>i} \hat{\mathbf{W}}_j'^{(t-1)}\right)$
13:         *// Apply SVID to find the best rank-1 approximation*
14:         $(\mathbf{B}_i'^{(t)}, \mathbf{g}_i'^{(t)}, \mathbf{h}_i'^{(t)}) \leftarrow \mathrm{SVID}(\mathbf{R}_i'^{(t)})$
15:         $\hat{\mathbf{W}}_i'^{(t)} \leftarrow \mathbf{g}_i'^{(t)} \odot \mathbf{B}_i'^{(t)} \odot \mathbf{h}_i'^{(t)}$
16:     **end for**
17: **end for**

18: *// Step 3: Map scales back to the original weight domain*
19: **for** $i = 1$ to $k$ **do**
20:     $\mathbf{g}_i \leftarrow \mathbf{s}_{\mathrm{out}}^{-\alpha_{\mathrm{out}}} \odot \mathbf{g}_i'^{(T_{\max})}$
21:     $\mathbf{h}_i \leftarrow \mathbf{s}_{\mathrm{in}}^{-\alpha_{\mathrm{in}}} \odot \mathbf{h}_i'^{(T_{\max})}$
22: **end for**

---

**Algorithm 2** RaBiT: Residual-Aware Binarization Training (One Step)

---

1: **Parameters:** Shared full-precision weight $\mathbf{W}_{\mathrm{FP}}$; Scales $\{(\mathbf{g}_i, \mathbf{h}_i)\}_{i=1}^k$.
2: **Input:** Minibatch Input $\mathbf{X}$, Targets $\mathbf{T}$.
3: *// 1. Forward Pass: On-the-fly Residual Coupling*
4: $\mathbf{R}_0 \leftarrow \mathbf{W}_{\mathrm{FP}}$.    *// Initialize residual with the shared weight*
5: $\hat{\mathbf{W}}^{(k)} \leftarrow \mathbf{0}$.    *// Effective weight for the entire layer*
6: **for** $i = 1$ to $k$ **do**
7:     *// Sequentially derive the i-th binary path*
8:     $\mathbf{B}_i \leftarrow \mathrm{sign}(\mathbf{R}_{i-1})$.
9:     $\hat{\mathbf{W}}_i \leftarrow \mathbf{g}_i \odot \mathbf{B}_i \odot \mathbf{h}_i$.
10:     $\hat{\mathbf{W}}^{(k)} \leftarrow \hat{\mathbf{W}}^{(k)} + \hat{\mathbf{W}}_i$.
11:     *// Update residual for the next path*
12:     $\mathbf{R}_i \leftarrow \mathbf{R}_{i-1} - \hat{\mathbf{W}}_i$.
13: **end for**
14: $\mathbf{Y} \leftarrow \hat{\mathbf{W}}^{(k)}\mathbf{X}$.    *// Compute layer output*
15: Calculate Loss $\mathcal{L}(\mathbf{Y}, \mathbf{T})$.
16: *// 2. Backward Pass*
17: $\Delta \leftarrow \partial\mathcal{L}/\partial\mathbf{Y}$.    *(Output gradient)*
18: *// Surrogate gradient for the shared weight $\mathbf{W}_{\mathrm{FP}}$*
19: $\nabla_{\mathbf{W}_{\mathrm{FP}}} \leftarrow \Delta^\top\mathbf{X}$.
20: *// Gradients for scales (treating $\mathbf{B}_i$ as constant)*
21: **for** $i = 1$ to $k$ **do**
22:     Compute $\nabla_{\mathbf{g}_i}$ and $\nabla_{\mathbf{h}_i}$ using $\Delta, \mathbf{B}_i, \mathbf{X}$, and other scales.
23: **end for**
24: *// 3. Parameter Update*
25: Update $\{\mathbf{W}_{\mathrm{FP}}, (\mathbf{g}_i, \mathbf{h}_i)_{i=1}^k\}$ using an optimizer with the computed gradients.

---

## A.10 Core Kernel Code

```
1   #include <ATen/cuda/CUDAContext.h>
2   #include <c10/cuda/CUDAException.h>
3   #include <c10/cuda/CUDAGuard.h>
4   #include <c10/macros/Macros.h>
5   #include <cuda_fp16.h>
6   #include <cuda_runtime.h>
7   #include <torch/extension.h>
8
9   #include <sstream>
10  #include <string>
11  #include <type_traits>
12
13  // ========================================================================
14  // === Macros & Constants ===
15  // ========================================================================
16
17  #define CHECK_CUDA(x) TORCH_CHECK(x.is_cuda(), #x " must be a CUDA tensor")
18  #define CHECK_CONTIGUOUS(x) TORCH_CHECK(x.is_contiguous(), #x " must be contiguous")
19  #define CHECK_F16(x) TORCH_CHECK(x.scalar_type() == torch::kFloat16, #x " must be float16 tensor")
20  #define CHECK_INT32(x) TORCH_CHECK(x.scalar_type() == torch::kInt32, #x " tensor dtype must be
        int32")
21
22  #define CHECK_CUDA_CONT_F16(x) \
23      do {                       \
24          CHECK_CUDA(x);         \
25          CHECK_CONTIGUOUS(x);   \
26          CHECK_F16(x);          \
27      } while (0)
28
29  #define CHECK_CUDA_CONT_INT32(x) \
30      do {                         \
31          CHECK_CUDA(x);           \
32          CHECK_CONTIGUOUS(x);     \
33          CHECK_INT32(x);          \
34      } while (0)
35
36  constexpr int WARP_SIZE = 32;
37  constexpr uint32_t SIGN_MASK_U32 = 0x80008000u;
38  constexpr unsigned FULL_MASK = 0xffffffff;
39
40  // ========================================================================
41  // === Device Helper Functions ===
42  // ========================================================================
43
44  static __device__ __forceinline__ uint32_t half2_to_uint32(const half2& h) {
45      return reinterpret_cast<const uint32_t&>(h);
46  }
47
48  static __device__ __forceinline__ half2 uint32_to_half2(uint32_t v) {
49      return reinterpret_cast<const half2&>(v);
50  }
51
52  // Sign-flipping utility.
53  // Logic: bits determine if we flip the sign of the float16 values packed in uint4.
54  __device__ __forceinline__ uint4 apply_sign(uint4 x4, uint32_t bits, int shift_base) {
55      uint32_t shifted = bits << shift_base;
56      x4.x ^= shifted & SIGN_MASK_U32;
57      x4.y ^= (shifted << 1) & SIGN_MASK_U32;
58      x4.z ^= (shifted << 2) & SIGN_MASK_U32;
59      x4.w ^= (shifted << 3) & SIGN_MASK_U32;
60      return x4;
61  }
62
63  // ========================================================================
64  // === Kernel Implementation ===
65  // ========================================================================
66
67  template <unsigned NUM_THREAD, unsigned NUM_ROW_PER_WARP = 2, unsigned NUM_ACC = 4, unsigned
        NUM_PIPELINE_STAGE = 2>
68  __forceinline__ __device__ void rabit_2bit_half_impl(
69      const half* __restrict__ x,
70      const half* __restrict__ scale_g_0,
71      const uint32_t* __restrict__ Wbits,
72      const half* __restrict__ scale_h_0,
73      const half* __restrict__ scale_g_1,
74      const half* __restrict__ scale_h_1,
75      half* __restrict__ y,
76      int N, int M) {
```

```
 77
 78    // Constraints
 79    static_assert(NUM_ROW_PER_WARP == 2, "Only 2 rows per warp supported currently");
 80    static_assert(NUM_THREAD % WARP_SIZE == 0, "Threads must be multiple of warp size");
 81
 82    constexpr unsigned NUM_WARP = NUM_THREAD / WARP_SIZE;
 83    constexpr unsigned NUM_ROW_PER_BLOCK = NUM_ROW_PER_WARP * NUM_WARP;
 84
 85    // Identifiers
 86    const int tid = threadIdx.x % WARP_SIZE;
 87    const int wid = threadIdx.x / WARP_SIZE;
 88
 89    // Vectorized pointers (loading 128-bit chunks / 8 halves at a time)
 90    const uint4* x8s = reinterpret_cast<const uint4*>(x);
 91    const uint4* sg8s_0 = reinterpret_cast<const uint4*>(scale_g_0);
 92    const uint4* sg8s_1 = reinterpret_cast<const uint4*>(scale_g_1);
 93
 94    // Data Structures for Pipelining
 95    struct RowAccum {
 96        half2 plane[2][NUM_ACC];
 97    };
 98    struct ColumnScalePair {
 99        uint4 g0;
100        uint4 g1;
101    };
102    struct StageTile {
103        uint4 x;
104        ColumnScalePair scale;
105        uint32_t bits;
106    };
107    struct RowContext {
108        int index;
109        RowAccum acc;
110    };
111
112    RowContext rows[NUM_ROW_PER_WARP] = {};
113    StageTile pipe[NUM_PIPELINE_STAGE];
114
115    // Setup Row Indices
116    const int block_row_base = blockIdx.x * NUM_ROW_PER_BLOCK;
117    #pragma unroll
118    for (unsigned irow = 0; irow < NUM_ROW_PER_WARP; ++irow) {
119        rows[irow].index = block_row_base + static_cast<int>(wid) * NUM_ROW_PER_WARP +
      static_cast<int>(irow);
120    }
121
122    // Weight Bit Indexing
123    // N8 = Number of 8-half chunks (128-bit words)
124    const int N8 = N >> 3;
125    const int wordsN = N8;
126    const int tile_stride = wordsN;
127    const int tile_idx = blockIdx.x * NUM_WARP + wid;
128    const uint32_t* tile_ptr = Wbits + tile_idx * tile_stride;
129
130    // --- Helper Lambdas ---
131
132    auto load_to_reg = [&](int ireg, int idx8) {
133        StageTile& stage = pipe[ireg];
134        stage.x = x8s[idx8];
135        stage.scale.g0 = sg8s_0[idx8];
136        stage.scale.g1 = sg8s_1[idx8];
137        stage.bits = *(tile_ptr + idx8);
138    };
139
140    // Accumulator indices for circular buffer usage or simple unrolling
141    const int MASK = NUM_ACC - 1;
142    auto fma = [&](const uint4& scale, const uint4& x, half2 acc[NUM_ACC]) {
143        acc[0] = __hfma2(uint32_to_half2(scale.x), uint32_to_half2(x.x), acc[0]);
144        acc[1 & MASK] = __hfma2(uint32_to_half2(scale.y), uint32_to_half2(x.y), acc[1 & MASK]);
145        acc[2 & MASK] = __hfma2(uint32_to_half2(scale.z), uint32_to_half2(x.z), acc[2 & MASK]);
146        acc[3 & MASK] = __hfma2(uint32_to_half2(scale.w), uint32_to_half2(x.w), acc[3 & MASK]);
147    };
148
149    auto calc_main = [&](int ireg) {
150        StageTile& stage = pipe[ireg];
151        #pragma unroll
152        for (unsigned irow = 0; irow < NUM_ROW_PER_WARP; ++irow) {
153            RowAccum& acc = rows[irow].acc;
154            // Apply sign logic based on bits (0 or 1 selects path)
155            // 0 + 8*irow and 4 + 8*irow are specific bit shifts for the packed format
156            fma(stage.scale.g0, apply_sign(stage.x, stage.bits, 0 + 8 * irow), acc.plane[0]);
```

```
157                   fma(stage.scale.g1, apply_sign(stage.x, stage.bits, 4 + 8 * irow), acc.plane[1]);
158               }
159           };
160
161           // --- Main Loop (Pipelined) ---
162
163           int idx_load = tid;
164           int idx_calc = tid;
165
166           // Prologue: Fill pipeline
167           #pragma unroll
168           for (int istg = 0; istg < NUM_PIPELINE_STAGE; ++istg) {
169               if (idx_load < N8) {
170                   load_to_reg(istg, idx_load);
171               }
172               idx_load += WARP_SIZE;
173           }
174
175           // Steady state
176           const int bound_mainloop = N8 - (NUM_PIPELINE_STAGE * WARP_SIZE);
177           while (idx_calc < bound_mainloop) {
178               #pragma unroll
179               for (int istg = 0; istg < NUM_PIPELINE_STAGE; ++istg) {
180                   calc_main(istg);
181               }
182               #pragma unroll
183               for (int istg = 0; istg < NUM_PIPELINE_STAGE; ++istg) {
184                   if (idx_load < N8) {
185                       load_to_reg(istg, idx_load);
186                   }
187                   idx_load += WARP_SIZE;
188                   idx_calc += WARP_SIZE;
189               }
190           }
191
192           // Epilogue: Drain pipeline
193           #pragma unroll
194           for (int istg = 0; istg < NUM_PIPELINE_STAGE; ++istg) {
195               if (idx_calc < N8) {
196                   calc_main(istg);
197                   idx_calc += WARP_SIZE;
198               }
199               if (idx_load < N8) {
200                   load_to_reg(istg, idx_load);
201                   idx_load += WARP_SIZE;
202               }
203           }
204
205           // --- Reduction & Output ---
206
207           auto warp_sum_half = [](half v) {
208               half2 h2 = __halves2half2(v, __float2half(0.0f));
209               #pragma unroll
210               for (int off = WARP_SIZE >> 1; off > 0; off >>= 1) {
211                   half2 other = __shfl_xor_sync(FULL_MASK, h2, off);
212                   h2 = __hadd2(h2, other);
213               }
214               return __low2half(h2);
215           };
216
217           #pragma unroll
218           for (unsigned irow = 0; irow < NUM_ROW_PER_WARP; ++irow) {
219               RowContext& row = rows[irow];
220               const int out_i = row.index;
221               RowAccum& acc = row.acc;
222
223               half2 lane_acc_0 = __float2half2_rn(0.0f);
224               half2 lane_acc_1 = __float2half2_rn(0.0f);
225
226               #pragma unroll
227               for (unsigned iacc = 0; iacc < NUM_ACC; ++iacc) {
228                   lane_acc_0 = __hadd2(lane_acc_0, acc.plane[0][iacc]);
229                   lane_acc_1 = __hadd2(lane_acc_1, acc.plane[1][iacc]);
230               }
231
232               // Horizontal sum within registers (half2 -> half)
233               half sum0_h = __hadd(__low2half(lane_acc_0), __high2half(lane_acc_0));
234               half sum1_h = __hadd(__low2half(lane_acc_1), __high2half(lane_acc_1));
235
236               // Apply final scaling factors (h0, h1) and combine paths
237               half lane_total_h = __hfma(sum0_h, scale_h_0[out_i], __hmul(sum1_h, scale_h_1[out_i]));
```

```
238
239            // Cross-lane reduction
240            half warp_sum_h = warp_sum_half(lane_total_h);
241
242            if (tid == 0) {
243                y[out_i] = warp_sum_h;
244            }
245        }
246  }
247
248  template <unsigned NUM_THREAD, unsigned NUM_ROW_PER_WARP = 2, unsigned NUM_ACC = 4, unsigned
             NUM_PIPELINE_STAGE = 2>
249  __launch_bounds__(NUM_THREAD) __global__ void rabit_2bit_half_dyn_kernel(
250        const half* __restrict__ x,
251        const half* __restrict__ scale_g_0,
252        const uint32_t* __restrict__ Wbits,
253        const half* __restrict__ scale_h_0,
254        const half* __restrict__ scale_g_1,
255        const half* __restrict__ scale_h_1,
256        half* __restrict__ y,
257        int N, int M) {
258
259        const int s = blockIdx.y;
260        // Offset for batch size (sequence length)
261        const half* x_ptr_for_this_seq = x + s * N;
262        half* y_ptr_for_this_seq = y + s * M;
263
264        rabit_2bit_half_impl<NUM_THREAD, NUM_ROW_PER_WARP, NUM_ACC, NUM_PIPELINE_STAGE>(
265            x_ptr_for_this_seq, scale_g_0, Wbits, scale_h_0, scale_g_1, scale_h_1, y_ptr_for_this_seq,
           N, M);
266  }
267
268  // ========================================================================
269  // === Template Dispatcher ===
270  // ========================================================================
271
272  // Helper to sweep through a compile-time list of values.
273  template <typename T, T... values>
274  struct parameter_sweep {
275        template <typename F, typename U>
276        void operator()(const char* name, U runtime_value, F&& func) const {
277            bool found = false;
278            // Fold expression to iterate over template values
279            (
280                [&]() {
281                    if (!found && runtime_value == values) {
282                        std::forward<decltype(func)>(func)(std::integral_constant<T, values>{});
283                        found = true;
284                    }
285                }(),
286                ...);
287
288            if (!found) {
289                std::stringstream ss;
290                bool is_first = true;
291                (((is_first ? ss : (ss << ", ")) << values, is_first = false), ...);
292                TORCH_CHECK(false, name, " should be one of [", ss.str(), "], but ", runtime_value, "
           was given");
293            }
294        }
295  };
296
297  // ========================================================================
298  // === C++ Interface ===
299  // ========================================================================
300
301  torch::Tensor rabit_2bit_half_dyn_forward(
302        torch::Tensor x,
303        torch::Tensor scale_g_0, torch::Tensor Wbits, torch::Tensor scale_h_0,
304        torch::Tensor scale_g_1, torch::Tensor scale_h_1,
305        int64_t num_thread, int64_t num_row_per_warp, int64_t num_acc, int64_t num_pipeline_stage) {
306
307        CHECK_CUDA_CONT_F16(x);
308        CHECK_CUDA_CONT_F16(scale_g_0);
309        CHECK_CUDA_CONT_INT32(Wbits);
310        CHECK_CUDA_CONT_F16(scale_h_0);
311        CHECK_CUDA_CONT_F16(scale_g_1);
312        CHECK_CUDA_CONT_F16(scale_h_1);
313
314        TORCH_CHECK(x.dim() == 1 || x.dim() == 2, "Dynamic Half: x must be 1D or 2D");
315
```

```
316        const int seqlen = x.dim() == 1 ? 1 : static_cast<int>(x.size(0));
317        const int N = static_cast<int>(x.size(-1));
318        const int M = static_cast<int>(scale_h_0.size(0));
319
320        // Validations
321        TORCH_CHECK(scale_g_0.size(0) == N, "Dynamic Half: scale_g_0 size mismatch");
322        TORCH_CHECK(scale_h_0.size(0) == M, "Dynamic Half: scale_h_0 size mismatch");
323        TORCH_CHECK(scale_g_1.size(0) == N, "Dynamic Half: scale_g_1 size mismatch");
324        TORCH_CHECK(scale_h_1.size(0) == M, "Dynamic Half: scale_h_1 size mismatch");
325        TORCH_CHECK((N % 32 == 0), "Dynamic Half: N must be multiple of 32");
326        TORCH_CHECK(num_row_per_warp > 0, "Dynamic Half: num_row_per_warp must be positive");
327
328        const int rows_per_warp = static_cast<int>(num_row_per_warp);
329        TORCH_CHECK(M % rows_per_warp == 0, "Dynamic Half: rows_per_warp must divide output size");
330        TORCH_CHECK(Wbits.dim() == 2 && Wbits.size(0) == M / rows_per_warp && Wbits.size(1) == N / 8,
331                    "Dynamic Half: Wbits shape mismatch");
332
333        auto y = torch::empty({(long long)seqlen, (long long)M}, x.options());
334
335        at::cuda::CUDAGuard guard(x.device());
336        auto stream = at::cuda::getCurrentCUDAStream();
337
338        // Template Sweepers
339        parameter_sweep<unsigned, 32u, 64u, 128u, 256u, 512u, 1024u> sweep_num_thread;
340        parameter_sweep<unsigned, 2> sweep_row_per_warp;
341        parameter_sweep<unsigned, 1, 2, 4> sweep_num_acc;
342        parameter_sweep<unsigned, 1, 2, 3, 4> sweep_pipeline_stage;
343
344        // Dispatcher
345        auto kernel_func = [=](auto num_thread_c, auto num_row_per_warp_c, auto num_acc_c, auto
             num_pipeline_stage_c) {
346            constexpr unsigned num_thread_v = num_thread_c.value;
347            constexpr unsigned num_row_per_warp_v = num_row_per_warp_c.value;
348            constexpr unsigned num_acc_v = num_acc_c.value;
349            constexpr unsigned pipeline_stage_v = num_pipeline_stage_c.value;
350
351            // Runtime check against compile time constant wrapper
352            if (num_row_per_warp_v != static_cast<unsigned>(rows_per_warp)) return;
353
354            constexpr unsigned NUM_ROW_PER_BLOCK = (num_thread_v / WARP_SIZE) * num_row_per_warp_v;
355            TORCH_CHECK(M % NUM_ROW_PER_BLOCK == 0, "Output size ", M, " must be div by block rows ",
             NUM_ROW_PER_BLOCK);
356
357            dim3 grid_dim(M / NUM_ROW_PER_BLOCK, seqlen);
358
359            rabit_2bit_half_dyn_kernel<num_thread_v, num_row_per_warp_v, num_acc_v, pipeline_stage_v>
360                <<<grid_dim, num_thread_v, 0, stream>>>(
361                    reinterpret_cast<const half*>(x.data_ptr<at::Half>()),
362                    reinterpret_cast<const half*>(scale_g_0.data_ptr<at::Half>()),
363                    reinterpret_cast<const uint32_t*>(Wbits.data_ptr<int32_t>()),
364                    reinterpret_cast<const half*>(scale_h_0.data_ptr<at::Half>()),
365                    reinterpret_cast<const half*>(scale_g_1.data_ptr<at::Half>()),
366                    reinterpret_cast<const half*>(scale_h_1.data_ptr<at::Half>()),
367                    reinterpret_cast<half*>(y.data_ptr<at::Half>()),
368                    N, M);
369            C10_CUDA_KERNEL_LAUNCH_CHECK();
370        };
371
372        // Execute Sweep
373        sweep_num_thread("num_thread", num_thread, [&](auto num_thread_c) {
374            sweep_row_per_warp("num_row_per_warp", num_row_per_warp, [&](auto num_row_per_warp_c) {
375                sweep_num_acc("num_acc", num_acc, [&](auto num_acc_c) {
376                    sweep_pipeline_stage("num_pipeline_stage", num_pipeline_stage, [&](auto
             num_pipeline_stage_c) {
377                        kernel_func(num_thread_c, num_row_per_warp_c, num_acc_c, num_pipeline_stage_c);
378                    });
379                });
380            });
381        });
382
383        if (x.dim() == 1) {
384            y.squeeze_(0);
385        }
386        return y;
387    }
```

## A.11 LLM Usage

LLM is used only for writing, editing, or formatting purposes and does not impact the core methodology, scientific rigorousness, or originality of the research.

