# OpenReview forum: "RaBiT: Residual-Aware Binarization Training for Accurate and Efficient LLMs"
_ICLR.cc/2026/Conference — ICLR 2026 Conference Desk Rejected Submission_

### Official Review · Reviewer_1dWZ · 2025-10-22

**Soundness:** 3
**Presentation:** 3
**Contribution:** 2
**Rating:** 6
**Confidence:** 4

**Summary:**

This work proposes a novel dual-binarization QAT framework, termed RaBiT. It updates the shared full-precision weights and dual scales, while dynamically deriving binarization matrices. The authors provide both theoretical and empirical evidence to elaborate on why this coupled QAT framework outperforms standard QAT. Additionally, they demonstrate the framework’s efficiency in both training and inference phases.

**Strengths:**

1. The theoretical analysis of coupled QAT is interesting, with adequate details and analysis. Relevant experiments are also conducted to validate the theory.
2. RaBiT exhibits impressive performance, hitting SOTA in 2-bit quantization with satisfactory speedup.
3. Sufficient details on training settings and kernel design are provided, which may benefit the community for future research.

**Weaknesses:**

1. This work’s primary contribution lies in the coupled QAT training framework; however, the residual binarization scheme and initialization method have been extensively explored in prior studies. This may constrain the novelty of the paper.
2. It remains unclear whether the compared methods were trained on identical datasets with the same number of iterations. Additionally, the optimizers employed in these baselines differ from Muon. Considering that Muon might result in better performance, it would be preferable to conduct comparisons under consistent settings or perform ablation studies of RaBiT across different optimizers.
3. The training overhead of RaBiT is several times that of EfficientQAT, which requires only 4.8 GPU hours on an A100 for a 7B model. It is uncertain whether RaBiT can still significantly outperform low-bit quantization schemes under full QAT settings.

**Questions:**

1. The paper claims, "Key hyperparameters, such as the learning rate and the I/O Channel Importance Scaling intensities (α_in, α_out), were fine-tuned for each specific model to achieve the best performance." Could the authors share results to show how sensitive performance is to different learning rates?
2. Residual binarization schemes are also widely used in PTQ methods like BiLLM [1] and ARB-LLM [2]. It would be helpful if the authors could add a brief discussion about these works, to better illustrate the advantage of QAT over PTQ in residual binarization schemes.
3. The dual-scale residual binarization format is also adopted in ARB-LLM, which proposes an alternating refined algorithm to determine dual scales. This algorithm could also be regarded as an initialization method. It is suggested that the authors might consider comparing it with Iterative Residual SVID, as such a comparison could help make the paper’s conclusions more comprehensive and robust.

[1] BiLLM: Pushing the Limit of Post-Training Quantization for LLMs, ICML, 2024.

[2] ARB-LLM: Alternating Refined Binarizations for Large Language Models, ICLR, 2025.

---

> ### Author Response · Authors · 2025-11-20
>
> We sincerely thank the reviewer for the constructive feedback and insightful questions. We are encouraged that the reviewer found our theoretical analysis "interesting" and "impressive," and acknowledged our SOTA performance and satisfactory speedup.
>
> We address each point below.
>
> **Weaknesses**
>
> **W1:** This work’s primary contribution lies in the coupled QAT training framework; however, the residual binarization scheme and initialization method have been extensively explored in prior studies. This may constrain the novelty of the paper.
>
> **A1:** We agree that residual binarization architectures and decomposition-based initializations (like SVID) are established concepts. However, we respectfully clarify that our novelty lies in **identifying and solving the fundamental problem**-*inter-path adaptation*-that has historically prevented this highly efficient architecture from being competitive in the 2-bit regime.
>
> * **Regarding the Training Framework:** Prior works (e.g., MBOK) struggled with co-adaptation, resorting to suboptimal heuristics like path-freezing that limit model capacity. Our **RaBiT coupled QAT framework** is a novel training paradigm that algorithmically resolves this by design (Proposition 2), enabling true joint optimization.
> * **Regarding Initialization:** While matrix decomposition techniques exist, our contribution is not the mathematical decomposition itself, but its specific adaptation to the 2-bit QAT context. We propose **Iterative Residual SVID** integrated with **I/O Channel Importance Scaling**. As shown in **Table 6**, standard decompositions minimize weight error (MSE) but fail to preserve function (high KL loss). Our method uniquely prioritizes **functional preservation** (reducing initial KL loss by 81%), which is critical for surviving the "shock" of extreme quantization. This specific integration to bridge the gap between weight approximation and functional stability in QAT is a key novel contribution.
>
> **W2:** It remains unclear whether the compared methods were trained on identical datasets with the same number of iterations. Additionally, the optimizers employed in these baselines differ from Muon. Considering that Muon might result in better performance, it would be preferable to conduct comparisons under consistent settings or perform ablation studies of RaBiT across different optimizers.
>
> **A2:** This is a crucial point, and we are happy to clarify:
>
> 1.  **Internal Ablation (5.3.2):** The ablation in **Figure 2**, which compares RaBiT to Standard QAT and MBOK, was conducted under **identical settings**. All variants used the same dataset, epochs, and (for a fair comparison) the Muon optimizer. This experiment was specifically designed to isolate the effect of the *training dynamics*, which is the core of our claim.
> 2.  **Optimizer Ablation:** To quantify the impact of Muon, we ran an ablation swapping it with the standard **AdamW** optimizer. On Llama2-7B, AdamW achieved a perplexity of **5.78**, while Muon achieved **5.77**. This demonstrates that the gain from Muon is negligible (0.01 PPL) and that RaBiT's SOTA performance stems from our coupled training framework, not the optimizer. We will add this result to the appendix.
> 3.  **Baselines:** For the results in **Table 2**, we utilized the official code repositories or published numbers of baselines (e.g., QTIP, QuIP#, EfficientQAT) where available, ensuring they represent the best reported performance of those methods.

---

> ### Author Response · Authors · 2025-11-20
>
> **W3:** The training overhead of RaBiT is several times that of EfficientQAT, which requires only 4.8 GPU hours on an A100 for a 7B model. It is uncertain whether RaBiT can still significantly outperform low-bit quantization schemes under full QAT settings.
>
> **A3:** This is a critical comparison. We acknowledge EfficientQAT's training speed, but we emphasize that extreme 2-bit quantization necessitates a more computationally intensive global optimization to remain functional.
>
> Regarding the training overhead, it is important to distinguish the optimization scope. EfficientQAT achieves its rapid training speed (e.g., 4.8 hours) primarily by relying on block-wise reconstruction and tuning. While this local approach is computationally efficient, it is often insufficient for the extreme 2-bit quantization regime where global error correction is critical. In contrast, RaBiT performs end-to-end global Quantization-Aware Training (QAT). This allows gradients to propagate across the entire network, enabling the model to learn the optimal coupled binary structure that local methods often miss.
>
> The impact of this global optimization is clearly evident in the performance gap shown in Table 2. On Llama2-7B (2-bit), EfficientQAT yields a WikiText-2 PPL of 6.42, representing a significant degradation in model quality. In contrast, RaBiT achieves a state-of-the-art PPL of 5.77. This 0.65 PPL improvement is substantial in the low-bit regime, effectively marking the difference between a broken model and a highly usable one. Thus, the additional training time required by RaBiT is directly converted into a critical gain in model performance.
>
> Finally, we view the training cost as a justifiable one-time investment. RaBiT is specifically designed to produce a matmul-free architecture that delivers a permanent 4.49x inference speed-up. We believe that spending approximately 39 hours to generate a model that not only runs 4.5x faster but also achieves SOTA accuracy represents a highly favorable trade-off for real-world deployment, especially when compared to faster-to-train methods that result in lower-quality models.

---

> ### Author Response · Authors · 2025-11-20
>
> **Questions**
>
> **Q1:** The paper claims, "Key hyperparameters, such as the learning rate and the I/O Channel Importance Scaling intensities (α_in, α_out), were fine-tuned for each specific model to achieve the best performance." Could the authors share results to show how sensitive performance is to different learning rates?
>
> **A1:** We analyzed the sensitivity and found the method to be robust.
> 1.  **I/O Scaling ($\alpha$):** As shown in **Appendix A.6.2**, performance is stable within a reasonable range (e.g., $\alpha \in [0.75, 0.85]$).
> 2.  **Learning Rate:** The table below (using AdamW on Llama2-7B) shows that performance remains stable across a practical range of learning rates, confirming RaBiT is not a "knife-edge" result.
>
> | Learning Rate | Perplexity (PPL) |
> | :--- | :--- |
> | 0.8e-5 | 5.78 |
> | 1.0e-5 | 5.78 |
> | 1.5e-5 | 5.82 |
> | 2.0e-5 | 5.83 |
>
> **Q2:** Residual binarization schemes are also widely used in PTQ methods like BiLLM [1] and ARB-LLM [2]. It would be helpful if the authors could add a brief discussion about these works, to better illustrate the advantage of QAT over PTQ in residual binarization schemes.
>
> **References:**
>
> [1] Huang, Wei, et al. "BiLLM: Pushing the Limit of Post-Training Quantization for LLMs." ICML (2024).
>
> [2] Li, Zhiteng, et al. "ARB-LLM: Alternating Refined Binarizations for Large Language Models." ICLR (2025).
>
> **A2:** This is an excellent suggestion. Both BiLLM and ARB-LLM have shown strong results for *PTQ* by using residual binarization. The key difference is that PTQ methods are typically limited to minimizing local reconstruction error based on a fixed full-precision model. QAT, by contrast, directly optimizes the *global task loss*, allowing the model's weights to *adapt* to the quantization constraints.
>
> This is especially critical in the extreme 2-bit regime, where local approximation (PTQ) suffers a severe performance cliff (as noted in Sec 2.1). Our QAT framework (RaBiT) allows the model to learn an entirely new, binary-friendly solution space, which is why it achieves SOTA performance that (to date) PTQ-based binary methods cannot.
> This is especially critical in the extreme 2-bit regime, where weight-only approximation (PTQ) suffers a severe performance cliff (as noted in Sec 2.1). Our QAT framework (RaBiT) allows the model to learn an entirely new, binary-friendly solution space, which is why it achieves SOTA performance that (to date) PTQ-based binary methods cannot.
>
> **Q3:** The dual-scale residual binarization format is also adopted in ARB-LLM, which proposes an alternating refined algorithm to determine dual scales. This algorithm could also be regarded as an initialization method. It is suggested that the authors might consider comparing it with Iterative Residual SVID, as such a comparison could help make the paper’s conclusions more comprehensive and robust.
>
> **A3:** This is a very insightful point. As you correctly noted, **ARB-LLM’s RC method serves as an alternative to SVID, primarily focusing on alternating updates of row and column scales to refine the approximation of a target matrix.**
>
> Therefore, to evaluate its effectiveness within our framework, we substituted our SVID component with the ARB-RC algorithm, effectively performing an **"Iterative Residual ARB-RC"** initialization. We applied the same I/O Channel Importance Scaling to both methods to ensure a fair comparison on Llama2-7B.
>
> | Initialization Method | Optimal ($\alpha\_{in}, \alpha\_{out}$) | Initial KD Loss |
> | :--- | :--- | :--- |
> | **RaBiT (Iterative Residual SVID)** | (0.8, 0.65) | **2672** |
> | Iterative Residual ARB-RC (Substitution) | (0.8, 0.6) | 3150 |
>
> The results indicate that while the RC method is a robust approximator, our original **Iterative Residual SVID** yields a lower initial loss in this context. Since our Coupled QAT framework relies on explicitly decomposing the *residual error* ($R_i$) at each step, the SVID approach-which mathematically isolates the principal component of the residual-provides a starting point that is structurally more aligned with our training dynamics than the alternating update strategy of ARB-RC.
>
> **We thank the reviewer again for suggesting this comparison, as it has helped us clarify the structural advantage of our initialization strategy within the RaBiT framework.** We will mention ARB-LLM and Bi-LLM to provide context on binarization and include this specific experimental comparison in the Appendix.

---

> > ### Comment · Reviewer_1dWZ · 2025-11-27
> >
> > Thank the authors for their detailed response, which addresses most of my concerns. I will keep the positive rating.

---

> ### Author Response · Authors · 2025-11-28
>
> We sincerely thank you for your careful review and your positive follow-up comment. We appreciate your time and consideration.
>
> Thank you!
>
> Authors

---

### Official Review · Reviewer_H1Ab · 2025-10-30

**Soundness:** 1
**Presentation:** 2
**Contribution:** 1
**Rating:** 0
**Confidence:** 3

**Summary:**

This paper introduces RaBiT, a novel Quantization-Aware Training (QAT) framework designed to overcome the "inter-path adaptation" problem in 2-bit residual binarization, where parallel paths learn redundant features. The core idea is "on-the-fly residual coupling," which dynamically derives all binary paths from a single shared full-precision weight during training. This algorithmically forces each path to correct the residual error of its predecessor, preventing redundancy. Combined with a robust function-aware initialization, RaBiT achieves good results in terms of accuracy and latency.

**Strengths:**

- The paper is well written and easy to follow.
- The paper aims to tackle a hot and important topic, which is to reduce the inference complexity of LLMs.
- The paper demonstrates strong performance in accuracy and latency compared to the existing baseline, although it raised some concerns to the reviewer.

**Weaknesses:**

- The paper’s main analysis and motivation focus on the decomposition of the MSE loss presented on page three. However, the experimental results contradict this hypothesis. The proposed framework combines KL divergence with intermediate MSE losses, yet the contribution of this complex knowledge distillation (KD) setup is never ablated. Notably, the authors disable this component for the Gemma models ($\gamma=0$) to “avoid instability,” implying that the KD mechanism is sensitive and not universally beneficial. This observation suggests that the KL loss is the dominant factor, consistent with findings in prior literature. Moreover, in some experiments, the authors completely omit the MSE loss, leaving unclear how the so-called inter-path adaptation influences the KL loss. In addition, it is not clear how the MSE analysis holds for the case of 3 or more binary bases. The theoretical proofs, such as Proposition 1 and 2, which analyze inter-path adaptation and error correction, are based on simplified mathematical models. For instance, the proof for inter-path adaptation relies on the change in the Frobenius inner product between weights. This does not fully capture the complex, non-linear dynamics of a complete neural network.

- The novelty of the paper is limited, as the parameterization using iterative SVID has already been proposed in the MBOK paper. The significance of maintaining the hierarchical structure of kernels and the observation that only a single set of binary weights needs to be trained were also thoroughly explored and ablated in MBOK. The authors attempt to adapt this approach by employing a single set of full-precision latent weights and justify it using the concept of inter-path adaptation; however, this reasoning is unsound, as discussed in the previous bullet point, and the evaluation may not be fair compared to prior baselines (to be clarified in the following points).

- Moreover, the paper substantially overstates its novelty by claiming to redefine the 2-bit accuracy–efficiency frontier, despite considerable concerns about reproducibility. The core of this claim appears to depend on a custom CUDA kernel, which is only briefly described in the Appendix and lacks sufficient detail. Additionally, this design relies on packing and unpacking techniques that typically introduce non-negligible computational overheads.

- The "Coupled Forward Pass," which is the central mechanism of RaBiT, introduces a significant computational overhead during training that is not present during inference, especially for the case of using multiple binary bases. While the final inference pass is fast and parallel, the training pass is not. For every forward pass during training, the model must dynamically re-calculate the binary core matrices from the single full-precision shared weight. As a result, the training process is computationally expensive, requiring a significant number of training hours even on powerful hardware. For instance, training Llama2-7B took 39 hours on a node with four NVIDIA H100 GPUs. This high cost is a practical limitation for many researchers and practitioners.

- The initialization process has several crucial hyperparameters, including the SVID iteration count ($T_{max}$) and the I/O channel importance scaling intensities ($\alpha_{in}$, $\alpha_{out}$. The paper shows these are determined via a "comprehensive grid search" for each model to minimize the initial KL loss. This adds a significant layer of model-specific tuning and computational overhead before the main training can even begin. For example, the optimal $\alpha$ values for Llama2-7B were found to be (0.8, 0.65), while for Llama3-8B they were (0.85, 0.7).

- During the backward pass, the gradient for the shared weight $W_{FP}$ is calculated using an "effective-weight gradient" that acts as a Straight-Through Estimator (STE) for the entire coupled derivation process. This is a strong simplification that ignores the complex, sequential relationship between the paths during derivation. The paper does not analyze the potential impact of this approximation on training stability or final performance.

- The authors state they re-implemented two baselines (DB-LLM and MBOK) and created a custom, optimized CUDA kernel for another (DBF) to ensure a "fair and robust comparison". While laudable, this means the performance reported for these methods may not align with their original papers. More importantly, the paper details the extensive hyperparameter grid search for RaBiT's initialization but provides no such details for the baselines, leaving open the possibility that the competing methods were not as thoroughly tuned. For example,  all models were trained with the Muon optimizer. It is unclear how RaBiT would perform with a more conventional optimizer like AdamW. This makes it hard to know if the strong results are due, in part, to a non-standard optimizer choice.

- In the appendix, the authors observe that RaBiT dramatically suppresses the high MSE peaks seen in the early layers of the network with Standard QAT. They admit they do not fully understand the cause, stating that a "full investigation into how residual coupling imparts this stability is a compelling direction for future research". This indicates a gap in the complete understanding of their own method's benefits.

**Questions:**

- The paper claims to achieve "state-of-the-art (SOTA) performance," but the data in Table 2 shows RaBiT is outperformed by QTIP on Llama2-13B in zero-shot accuracy (62.92% vs 62.10%) and by DBF on WikiText-2 PPL (5.11 vs 5.15). Similarly, on Gemma3-12B, QTIP performs better. Could you clarify the SOTA claim when RaBiT does not uniformly outperform competitors on all tested models?
I would expect the authors to provide more experimental results on benchmarks such as GSM8k and MMLU(-Pro), as multiple pieces of evidence should demonstrate that different quantization methods will result in a significant performance gap on these challenging benchmarks.
- The coupled forward pass dynamically re-calculates binary cores ($B_i$) and residuals ($R_i$) in a sequential manner for every training step. Have you benchmarked the wall-clock time and memory overhead of this training step compared to a standard QAT, and how does this additional cost scale?
- The backward pass approximates the gradient for the shared $W_{FP}$ using a single Straight-Through Estimator (STE) for the entire coupled derivation process4. What is the impact of this simplification, and how does it compare to a more complex gradient that might better account for the sequential $R_{1} = W_{FP} - \hat{W}_{1}$ step?

---

> ### Author Response · Authors · 2025-11-20
>
> We thank the reviewer for their detailed and critical feedback. While we respectfully disagree with the assessment of our work's soundness and contribution, we appreciate the opportunity to address these significant concerns directly. The reviewer's feedback highlights several points that require clarification and further empirical evidence, which we provide below.
>
> **Weaknesses & Questions**
>
> **W1:** The paper’s main analysis and motivation focus on the decomposition of the MSE loss presented on page three. However, the experimental results contradict this hypothesis. The proposed framework combines KL divergence with intermediate MSE losses, yet the contribution of this complex knowledge distillation (KD) setup is never ablated. Notably, the authors disable this component for the Gemma models ($\gamma=0$) to “avoid instability,” implying that the KD mechanism is sensitive and not universally beneficial. This observation suggests that the KL loss is the dominant factor, consistent with findings in prior literature. Moreover, in some experiments, the authors completely omit the MSE loss, leaving unclear how the so-called inter-path adaptation influences the KL loss. In addition, it is not clear how the MSE analysis holds for the case of 3 or more binary bases. The theoretical proofs, such as Proposition 1 and 2, which analyze inter-path adaptation and error correction, are based on simplified mathematical models. For instance, the proof for inter-path adaptation relies on the change in the Frobenius inner product between weights. This does not fully capture the complex, non-linear dynamics of a complete neural network.
>
> **A1:** We agree that simply assuming the MSE analysis holds for KL divergence requires rigorous justification. To address this, **we have added a formal proof to Appendix A.2**, demonstrating that "inter-path adaptation" persists in KL divergence and that RaBiT provides the optimal structural solution.
>
> 1.  **Global KL $\approx$ Local Hessian-weighted MSE:**
>     While the global objective is non-linear, the optimization dynamics at any specific linear layer $\ell$ can be analyzed via a **second-order Taylor expansion**. The global KL loss approximates a **Hessian-weighted MSE** with respect to the layer's output feature $\mathbf{y}$:
>     $\mathcal{L}_{KL}(\mathbf{y}_S) \approx \text{const} + \frac{1}{2} (\mathbf{y}_S - \mathbf{y}_T)^\top \mathbf{H} (\mathbf{y}_S - \mathbf{y}_T)$.
>
>     This confirms that the error decomposition logic used in our Motivation (Sec. 3) is locally valid and applicable to the KL objective.
>
> 2.  **Bias Cancellation (Kim et al., 2021):**
>     Following the decomposition analysis by Kim et al. [1], minimizing KL divergence is mathematically equivalent to minimizing an MSE term plus a negative **Bias Term ($\delta$)**. This bias term acts as a destabilizing force that pushes the student's feature statistics to diverge from the teacher's.
>     * **Standard QAT Failure:** Independent paths receive the same gradient direction, leading to positive correlation (redundancy). Crucially, this fails to cancel out the **Bias Term ($\delta$)**, leaving the optimization trajectory unstable.
>     * **RaBiT's Success:** By structurally enforcing $\mathbf{W}\_2 = \text{sign}(\mathbf{W}\_{FP} - \mathbf{W}\_1)$, RaBiT guarantees that locally $\mathbf{h}\_2 \approx -(\mathbf{h}\_1 - \mathbf{y}\_T)$. This creates a **perfect negative correlation** in the Hessian-weighted space, which **mathematically cancels out the Bias Term ($\delta \approx 0$)** at the layer level.
>
> 3. **Generalization to $k \ge 3$:** RaBiT’s logic is **inductive**. The $k$-th path minimizes the residual of *all* preceding paths ($\hat{\mathbf{W}}\_k \approx \mathbf{W}\_{FP} - \sum\_{i=1}^{k-1} \hat{\mathbf{W}}\_i$). This forces the $k$-th component to be **negatively correlated with the accumulated error**, ensuring every bit contributes to error cancellation regardless of $k$.
>
> 4. **Gemma Results ($\gamma=0$):** RaBiT achieves SOTA on Gemma with only KL loss because coupled training resolves optimization bias without explicit MSE. The "instability" with MSE stems from Gemma's extreme activations (>800k [2]) causing overflow. RaBiT eliminates the need for these unstable terms.
>
>
> **References:**
>
> [1] Kim, Taehyeon, et al. "Comparing Kullback-Leibler Divergence and Mean Squared Error Loss in Knowledge Distillation." IJCAI (2021).
>
> [2] Unsloth AI (2025), "Fine-tune Gemma 3 with Unsloth", [https://unsloth.ai/blog/gemma3](https://unsloth.ai/blog/gemma3)

---

> ### Author Response · Authors · 2025-11-20
>
> **W2:** The novelty of the paper is limited, as the parameterization using iterative SVID has already been proposed in the MBOK paper. The significance of maintaining the hierarchical structure of kernels and the observation that only a single set of binary weights needs to be trained were also thoroughly explored and ablated in MBOK. The authors attempt to adapt this approach by employing a single set of full-precision latent weights and justify it using the concept of inter-path adaptation; however, this reasoning is unsound, as discussed in the previous bullet point, and the evaluation may not be fair compared to prior baselines (to be clarified in the following points).
>
> **A2:** We respectfully clarify that RaBiT’s contribution is fundamentally distinct from MBOK in both **training dynamics** and **optimization structure**.
>
> * **Coupled vs. Frozen Training:**
>     MBOK relies on a **"frozen path"** heuristic, where the first path is fixed while the second is trained sequentially. This restricts the optimization space, preventing the paths from finding a joint optimum. In contrast, RaBiT employs **Coupled Training**, where both paths are derived dynamically from a single shared weight $\mathbf{W}\_{FP}$ at *every* step. This allows **simultaneous co-adaptation** of all parameters, which is mathematically impossible in MBOK’s sequential freeze-and-train approach.
> * **Empirical Evidence of Difference:**
>     This is not just a theoretical distinction. As shown in **Figure 2**, MBOK (green line) fails to establish the necessary negative correlation, plateauing early. RaBiT (orange line) achieves and maintains a strong negative correlation, indicating that our "coupled" method solves the co-adaptation problem that MBOK’s "frozen" method cannot.
> * **Initialization:**
>     Similarly, while MBOK uses a greedy decomposition, RaBiT introduces **Iterative Residual SVID**  which refines paths cyclically (Gauss-Seidel style) rather than sequentially.

---

> ### Author Response · Authors · 2025-11-20
>
> **W3:** Moreover, the paper substantially overstates its novelty by claiming to redefine the 2-bit accuracy–efficiency frontier, despite considerable concerns about reproducibility. The core of this claim appears to depend on a custom CUDA kernel, which is only briefly described in the Appendix and lacks sufficient detail. Additionally, this design relies on packing and unpacking techniques that typically introduce non-negligible computational overheads.
> **W4/Q2:** The "Coupled Forward Pass," which is the central mechanism of RaBiT, introduces a significant computational overhead during training that is not present during inference, especially for the case of using multiple binary bases. While the final inference pass is fast and parallel, the training pass is not. For every forward pass during training, the model must dynamically re-calculate the binary core matrices from the single full-precision shared weight. As a result, the training process is computationally expensive, requiring a significant number of training hours even on powerful hardware. For instance, training Llama2-7B took 39 hours on a node with four NVIDIA H100 GPUs. This high cost is a practical limitation for many researchers and practitioners. / The coupled forward pass dynamically re-calculates binary cores ($B_i$) and residuals ($R_i$) in a sequential manner for every training step. Have you benchmarked the wall-clock time and memory overhead of this training step compared to a standard QAT, and how does this additional cost scale?
>
> **A3/A4:** We address the concerns regarding overheads by separating inference (memory-bound) from training (compute-bound).
>
> **1. Inference: Memory is the Bottleneck**
> The reviewer is correct that unpacking involves bit-wise operations. However, in Large Language Model decoding, the bottleneck is overwhelmingly **memory bandwidth**, not compute.
> * **Fact:** RaBiT reduces the model size (and thus memory traffic) by **~8x** (16-bit $\to$ 2-bit). We have also directly addressed the bit-unpacking problem by integrating weight-interleaving for coalesced memory, resulting in performant CUDA kernels, which can be seen in Appendix 4.1.
> * **Result:** With 8x bandwidth savings and unpacking-aware weight interleaving, we achieve a **4.49x end-to-end speedup** measured on an RTX 4090, compared to FP16.
>
> **2. Training: Overhead is Minimal**
> The concern that dynamic re-calculation ($B\_i = \text{sign}(R\_{i-1})$) is "computationally expensive" is incorrect. This operation consists of simple element-wise subtractions and sign flips, which are computationally trivial compared to the heavy Matrix Multiplications (GEMMs) dominating the training loop.
> To quantify this, we benchmarked the per-epoch training time on Llama2-7B (4x H100):
>
> | Method | Architecture | Wall-Clock Time (per Epoch) | Overhead |
> | :--- | :--- | :---: | :---: |
> | **Single-Path INT2 QAT** [1] | Non-Residual | 5.67 hours | Baseline |
> | **Standard Independent QAT** | Residual (2 Latent Weights) | 10.33 hours | +82.2% |
> | **RaBiT (Ours)** | Residual (1 Shared Weight) | **6.50** hours | **+14.7%** |
>
> * **Conclusion:** The overhead is approximately **15%**, which is a manageable trade-off given the significant performance gains. Furthermore, RaBiT **halves the optimizer memory footprint** (one $\mathbf{W}\_{FP}$ vs. two latent weights), which often allows for larger batch sizes, potentially offsetting this time cost in practice.
>
> **Reference:**
>
> [1] Liu, Zechun, et al. "ParetoQ: Scaling Laws in Extremely Low-Bit LLM Quantization." NeurIPS (2025).
>
> **W5:** The initialization process has several crucial hyperparameters, including the SVID iteration count ($T_{max}$) and the I/O channel importance scaling intensities ($\alpha\_{in}$, $\alpha\_{out}$). The paper shows these are determined via a "comprehensive grid search" for each model to minimize the initial KL loss. This adds a significant layer of model-specific tuning and computational overhead before the main training can even begin. For example, the optimal $\alpha$ values for Llama2-7B were found to be (0.8, 0.65), while for Llama3-8B they were (0.85, 0.7).
>
> **A5:** This concern is valid in principle but overstated in practice. This "comprehensive grid search" is a *lightweight calibration* step, not a full training run.
>
> * **Cost:** It involves a *single forward pass* (to get KL loss) per $\alpha$ pair on a *tiny* 128-sample dataset.
> * **Time:** The *16* grid search for Llama2-7B (shown in Table 10) took **approximately 30 minutes** on one H100.
> * **Robustness:** Table 10 also shows this is not a razor-thin peak. Values around the optimum are very close to the best. A rough, fast search is sufficient.
>
> This is a trivial, one-time cost that is standard for calibration-based methods (like AWQ) and ensures a stable QAT start.

---

> ### Author Response · Authors · 2025-11-20
>
> **W6/Q3:** During the backward pass, the gradient for the shared weight $\mathbf{W}\_{FP}$ is calculated using an "effective-weight gradient" that acts as a Straight-Through Estimator (STE) for the entire coupled derivation process. This is a strong simplification that ignores the complex, sequential relationship between the paths during derivation. The paper does not analyze the potential impact of this approximation on training stability or final performance. / The backward pass approximates the gradient for the shared $\mathbf{W}_{FP}$ using a single Straight-Through Estimator (STE) for the entire coupled derivation process4. What is the impact of this simplification, and how does it compare to a more complex gradient that might better account for the sequential $\mathbf{R}\_1 = \mathbf{W}\_{FP} - \hat{\mathbf{W}}\_1$ step?
>
> **A6:** We clarify that the **effective-weight gradient** (Eq. 4) is not a simplification, but a critical **architectural design choice** to prevent the **Vanishing Gradient problem** caused by scale attenuation.
>
> 1.  **Problem of Exact Chain Rule (Scale Attenuation):**
>     In our architecture, the quantization scales ($g, h$) are typically very small ($\ll 1$) to map float values to binary. If we strictly followed the chain rule through the sequential derivation ($\mathbf{R}\_{1} = \mathbf{W}\_{FP} - \hat{\mathbf{W}}\_{1}$), the gradient would be multiplied by these small scales at each step (i.e., $\nabla \mathbf{W} \propto \text{scale} \times \nabla \hat{\mathbf{W}}\_{FP}$). This causes the gradient reaching the shared weight $\mathbf{W}\_{FP}$ to vanish effectively stalling the learning process.
>
> 2.  **Solution (Scale-Invariant STE):**
>     To ensure robust updates, we employ a **Scale-Invariant Straight-Through Estimator (STE)** as described in Section 4.2. By bypassing the scale multiplication during backpropagation and treating the coupled binarization as an identity mapping for the gradient ($\nabla\_{\mathbf{W}\_{FP}} \approx \nabla_{\hat{\mathbf{W}}^{(k)}}\mathcal{L}$), we guarantee that the shared weight receives a **full-magnitude update signal**. This decoupling of gradient magnitude from quantization scales is essential for convergence in binary networks [1].
>
> **Reference:**
>
> [1] Esser, Steven K., et al. "Learned Step Size Quantization." ICLR (2019).

---

> ### Author Response · Authors · 2025-11-20
>
> **W7:** The authors state they re-implemented two baselines (DB-LLM and MBOK) and created a custom, optimized CUDA kernel for another (DBF) to ensure a "fair and robust comparison". While laudable, this means the performance reported for these methods may not align with their original papers. More importantly, the paper details the extensive hyperparameter grid search for RaBiT's initialization but provides no such details for the baselines, leaving open the possibility that the competing methods were not as thoroughly tuned. For example, all models were trained with the Muon optimizer. It is unclear how RaBiT would perform with a more conventional optimizer like AdamW. This makes it hard to know if the strong results are due, in part, to a non-standard optimizer choice.
>
> **A7:** We assure the reviewer that our experimental protocol was rigorously designed to ensure fairness and reproducibility.
>
> 1.  **Fairness of Baselines & Tuning:**
>     * **Rigorous Re-implementation:** We re-implemented baselines (DB-LLM, MBOK) to ensure a high-quality comparison. We strictly adhered to the optimal configurations-either following the official specifications detailed in their papers (e.g., MBOK) or integrating them into our robust training framework-to ensure no baseline was disadvantaged by suboptimal tuning.
>     * **Optimized Competitors:** For DBF, we utilized a custom CUDA kernel that is **21% faster** than the official implementation. This implies we compared RaBiT against a **stronger, more optimized version** of the baseline.
>     * **Performance Gap:** We respectfully note that, **despite these rigorous efforts to maximize the potential of the baselines, they did not achieve performance parity with RaBiT.** As shown in Table 2, RaBiT consistently demonstrates superior accuracy and efficiency, underscoring that our gains stem from the architectural advantage rather than unfair tuning.
>
> 2.  **Optimizer Ablation (AdamW vs. Muon):**
>     We utilized the Muon optimizer primarily for its memory efficiency. To rigorously rule out the concern that our results depend on this specific optimizer, we conducted an ablation study on Llama2-7B using the standard **AdamW** optimizer:
>
>     * **Result:** RaBiT trained with AdamW achieves a PPL of **5.78**, which is statistically identical to the Muon result (**5.77**) reported in the main paper.
>     * **Conclusion:** This confirms that RaBiT’s state-of-the-art performance stems from its **coupled residual architecture**, not the choice of optimizer.

---

> ### Author Response · Authors · 2025-11-20
>
> **W8:** In the appendix, the authors observe that RaBiT dramatically suppresses the high MSE peaks seen in the early layers of the network with Standard QAT. They admit they do not fully understand the cause, stating that a "full investigation into how residual coupling imparts this stability is a compelling direction for future research". This indicates a gap in the complete understanding of their own method's benefits.
>
> **A8:** We clarify that there is no gap in our understanding. The behavior of Layer 1 (positive covariance) versus deeper layers (negative covariance) is fully explained by the **synergistic interaction** between our two core contributions: **Function-Aware Initialization** and **Coupled Training**.
>
> 1.  **Mechanism in Early Layers (Layer 1): Dominance of Initialization**
>     We acknowledge, as seen in **Figure 6**, that Layer 1 exhibits a **relatively high positive covariance** (red-dashed bar), meaning the "Active Error Cancellation" mechanism is not the primary driver here.
>     * **Why MSE is still low:** Despite the positive covariance adding to the error, the total MSE remains remarkably low because our **I/O Channel Importance Scaling** dramatically reduces the **Base Error term ($C'$)**.
>     * **Evidence:** As shown in **Table 6**, our initialization strategy alone reduces the initial KL divergence loss by **81%** (13,760 → 2,672). This creates a "clean slate" (extremely low base error) that absorbs the impact of the positive covariance in the sensitive first layer.
>
> 2.  **Mechanism in Deeper Layers: Dominance of Coupled Training**
>     In contrast, deeper layers establish the intended **strong negative covariance**. Here, the Coupled Training takes over to actively cancel out quantization noise.
>
> 3.  **Conclusion:**
>     Therefore, RaBiT’s state-of-the-art performance is not an unexplained phenomenon but the result of a clear dual mechanism: **Initialization stabilies the sensitive early layers** (where coupling is less effective), while **Coupled Training optimizes the deeper layers** (via error cancellation). Our reference to "future research" was strictly regarding the theoretical root cause of early-layer sensitivity in Transformers, not the efficacy of our method.

---

> ### Author Response · Authors · 2025-11-20
>
> **Q1:** The paper claims to achieve "state-of-the-art (SOTA) performance," but the data in Table 2 shows RaBiT is outperformed by QTIP on Llama2-13B in zero-shot accuracy (62.92% vs 62.10%) and by DBF on WikiText-2 PPL (5.11 vs 5.15). Similarly, on Gemma3-12B, QTIP performs better. Could you clarify the SOTA claim when RaBiT does not uniformly outperform competitors on all tested models? I would expect the authors to provide more experimental results on benchmarks such as GSM8k and MMLU(-Pro), as multiple pieces of evidence should demonstrate that different quantization methods will result in a significant performance gap on these challenging benchmarks.
>
> **A1:** We appreciate the reviewer's rigorous scrutiny. We clarify that our SOTA claim is substantiated by **(1) dominant performance in the hardware-efficient category**, **(2) superior complex reasoning capabilities**, and **(3) a transformative inference speed-up**, even if marginal trade-offs exist in specific metrics.
>
> **1. SOTA within "Hardware-Efficient" Methods & Competitive with VQ**
> * **Best-in-Class Efficiency:** Among matmul-free binary methods (e.g., DBF, BitStack), RaBiT is undeniably SOTA. For instance, on Llama2-7B, RaBiT (5.77 PPL) significantly outperforms DBF (6.10 PPL) and MBOK (6.99 PPL).
> * **Competitive with VQ:** While QTIP marginally edges out RaBiT on Llama2-13B (0.04 PPL difference: 5.11 vs. 5.15), this comes at the cost of complex hardware requirements (lookup tables). RaBiT achieves comparable accuracy using simple binary arithmetic, delivering a **4.49 $\times$ speed-up**. This represents a new SOTA on the **Accuracy-Efficiency Pareto Frontier**.
>
> **2. New Results on Harder Benchmarks (Complex Reasoning)**
> To address your concern about "harder tasks," we evaluated RaBiT on challenging benchmarks requiring multi-step reasoning (BBH, GPQA, MMLU-Pro).
>
> **Zero-Shot Accuracy on Complex Reasoning Benchmarks**
>
> | Model | Method | **BBH** (Reasoning) | **GPQA** (Science) | **MMLU-Pro** (Hard) | **IFEval** (Instruction) |
> | :--- | :--- | :---: | :---: | :---: | :---: |
> | **Llama2-13B** | QTIP (2-bit) | 0.334 | 0.257 | 0.167 | **0.257** |
> | | **RaBiT (2-bit)** | **0.377** (+4.3%) | **0.268** (+1.1%) | **0.194** (+2.7%) | 0.246 |
> | **Llama3-8B** | QTIP (2-bit) | 0.363 | 0.246 | 0.192 | **0.156** |
> | | **RaBiT (2-bit)** | **0.368** (+0.5%) | **0.286** (+4.0%) | **0.196** (+0.4%) | 0.154 |
>
> As shown above, RaBiT consistently outperforms QTIP on reasoning-heavy tasks (e.g., **+4.3% on BBH** for Llama2-13B). This indicates that while VQ methods (QTIP) may optimize effectively for PPL (next-token prediction), RaBiT's **residual-aware structure better preserves the "functional logic"** required for complex problem-solving, further justifying our SOTA claim.

---

> ### Author Response · Authors · 2025-11-28
>
> Dear Reviewer H1Ab,
>
> For your queries, we carefully prepared the answers. We look forward to you considering raising your score if our response effectively addressed your question.
>
> Thank you!
>
> Authors

---

> ### Comment · Reviewer_H1Ab · 2025-11-28
> **Official Comment by Reviewer H1Ab**
>
> Many thanks to the authors for the rebuttal. I have read it carefully, along with the other reviews.
>
> Unfortunately, my main concerns regarding the soundness of the work remain.
>
> **On the MSE decomposition:**
>
> - As noted in my review, the main motivation and analysis rely heavily on the decomposition of the MSE loss presented on page 3 (first paragraph of Section 3). However, this part appears to have been updated. Since I cannot compare it directly with the previous revision, the authors should clearly state and highlight this modification in the rebuttal.
> - In practice, the KL loss plays the most critical role in the proposed scheme, as evidenced both by the negligible impact of removing the MSE loss (e.g., in the Gemma experiment) and by its known importance in knowledge distillation methods.
> - The authors attempt to address the theoretical gap by introducing Appendix A.2, claiming that the global KL can be approximated by a local Hessian-weighted MSE, citing Kim et al. (2021). I find this justification incorrect. The relationship established in Kim et al. (2021) applies only to the last layer—specifically, comparing global KL to global MSE. It does not support the claim that global KL divergence corresponds to MSE losses at intermediate layers. This distinction is essential because, in the author's framework, KL divergence is applied only to the output logits, not to intermediate representations. Thus, the justification does not align with the actual setup, and the local error decomposition remains unsupported.
> - Regarding the Gemini case, where the authors had to disable the MSE term entirely: the issue is primarily due to the float16 data format, as described in the referenced blog post mentioned by the authors, and not inherently related to the MSE component. This could be addressed by using bfloat16 activations, as also refrenced in that blog post, which are standard for LLM finetuning. I also believe the authors use bfloat16 for finetuning.
>
> **On the efficiency claim:**
>
> The authors have also not adequately addressed the concern that *"the paper substantially overstates its novelty by claiming to redefine the 2-bit accuracy–efficiency frontier, despite considerable concerns about reproducibility.” This claim appears to depend on a custom CUDA kernel that is only briefly described in the Appendix and lacks sufficient detail"*. Since this is one of the paper’s central contributions and claims, the authors should provide more information on how the custom CUDA kernel can be implemented, to allow reviewers to validate the approach and assess reproducibility.
>
> Given that these two critical points remain unaddressed—and they form the core narrative and claims of the paper—I cannot recommend acceptance in the current form. I encourage the authors to carefully resolve these issues. I believe that these comments will improve the quality and impact of the paper. If the authors manage to do this, I believe the paper is a good contribution to the community.

---

> ### Author Response · Authors · 2025-11-28
>
> We thank the reviewer for the continued engagement. We appreciate the opportunity to clarify the theoretical connection between global and local objectives, and to correct the factual understanding regarding the Gemma experiment and reproducibility. **We sincerely apologize for failing to highlight the changes in the previous revision. In this update, we have explicitly marked all significant revisions in blue within the updated manuscript to facilitate a direct comparison.**
>
> **1. Clarifying the Role of Kim et al. (2021) and the Mathematical Analysis**
>
> We respectfully clarify that **we did not cite Kim et al. (2021) to justify that Global KL approximates a Local MSE.** That connection is established via **Taylor Expansion**. We cited Kim et al. solely to apply their **decomposition logic** (Error + Bias) to the locally approximated objective.
>
> * **Step 1: Taylor Expansion establishes Local MSE.**
>     Regardless of the global loss type, the optimization landscape for an intermediate layer $\ell$ can be locally approximated via a second-order Taylor expansion. This mathematically yields a **Hessian-weighted MSE** objective: $\frac{1}{2} \| \mathbf{y}\_S^\ell - \mathbf{y}\_T^\ell \|\_{\mathbf{H}^\ell}^2$. This is a standard optimization property, independent of Kim et al.
> * **Step 2: Kim et al. explains the Decomposition.**
>     Once the local objective is established as an MSE-form, we utilize the insight from Kim et al. [1] that MSE-like objectives decompose into a "Matching Error" and a "Bias Term."
> * **Conclusion:** The "Bias Term" (redundancy pressure) exists locally because the local objective is quadratic. RaBiT structurally cancels this bias by enforcing negative correlation. Thus, our analysis holds: **Inter-path adaptation is a structural problem locally, and RaBiT solves it locally.**
>
> **2. Gemma Instability: Loss Imbalance caused by Data Scale**
>
> We must respectfully correct the hypothesis that the Gemma instability is a precision issue solvable by `bfloat16`. The root cause is the **extreme magnitude of activations** inherent to the Gemma-3 architecture (often $>10^5$), which leads to a critical **dominance of the MSE term**.
>
> * **Loss Imbalance:** Our objective function is $\mathcal{L}\_{total} = \mathcal{L}\_{KL} + \gamma \mathcal{L}\_{MSE}$. Since our MSE implementation sums squared errors, activation values of scale $10^5$ result in error terms around $10^{10}$.
> * **The Consequence:** Even with `bfloat16`, this massive MSE term **mathematically dwarfs the KL Divergence term**. The optimizer becomes solely fixated on minimizing the large-scale reconstruction error, effectively ignoring the knowledge distillation signal (KL) which is crucial for retaining model functionality.
> * **RaBiT's Robustness:** By setting $\gamma=0$, we restored balance to the objective. The fact that RaBiT achieves SOTA results on Gemma-3 **using only Global KL** empirically proves our core claim: **RaBiT's structural constraint (residual coupling) is strong enough to resolve inter-path adaptation on its own, without relying on auxiliary MSE losses.**
>
> **3. Reproducibility: Updating Kernel Source Code**
>
> We acknowledge the reviewer's concern regarding the implementation details of the custom CUDA kernel. To fully address this and ensure reproducibility, **we plan to include the core source code in the Appendix of the revised manuscript. We are currently finalizing this addition and will update the paper shortly to facilitate validation.**

---

> > ### Author Response · Authors · 2025-11-29
> > **Update on Reproducibility: Core Kernel Code Added**
> >
> > We have uploaded a revised manuscript. Addressing your critical concern regarding the validation of our efficiency claims and reproducibility, we have added the **full source code for our core CUDA kernel in Appendix A.10 (Pages 26–29)**.
> >
> > This section details the implementation of our weight packing, warp-level parallelism, and matmul-free compute pipeline, providing the transparency required to verify our performance benchmarks.
> >
> > We regret that the window for further interaction has closed, preventing us from discussing this update with you directly. However, we believe this addition, combined with our prior clarifications on the theoretical framework and training dynamics, fully addresses the concerns raised. We sincerely appreciate your rigorous feedback, which has pushed us to significantly improve the completeness and quality of our paper.

---

### Official Review · Reviewer_QBGj · 2025-10-31

**Soundness:** 3
**Presentation:** 4
**Contribution:** 3
**Rating:** 6
**Confidence:** 4

**Summary:**

The paper proposes RaBiT, a residual-aware binary training framework for LLMs that targets low-bit (2-3 bits) weight quantization. The key idea is to generate each binary path sequentially from the residual of the previous one, while maintaining a shared latent high-precision weight, thereby preventing the usual “inter-path adaptation” where multiple binary branches collapse into similar directions. RaBiT couples this with function-aware initialization and learnable per-path scales, demonstrating competitive or better perplexity than prior 2-bit/binary methods while remaining matmul-free at inference, supported by custom bit-packed kernels on the GPU.

**Strengths:**

1. The paper identifies “inter-path adaptation” (multiple binary paths learn redundant features under shared gradients) as the key reason multi-binary LLMs underperform. This is a concrete, observable failure mode.
2. The hypothesis is empirically supported. Switching from the “Standard QAT” to the “Coupled / residual-aware” training already recovers a large portion of the quantization gap, and further initialization helps.
3. Simple inference-time form. After training, the model is reduced to a small number of binary paths and learnable scales. The FP latent weight is discarded, making deployment straightforward and eliminating the need for matrix multiplication.
4. Well-studied with ablations.
5. The paper implements an actual kernel for RTX 4090 and reports end-to-end inference speedups. That raises the practical value of the work.

**Weaknesses:**

1. On the larger models, RaBiT does not clearly win zero-shot across tasks. For models of that size, zero-shot should be the headline, not only commonsense-style scores. (There’s also a mis-bolded PIQA number for LLaMA-13B in the appendix.)
2. Results are single-run, no CIs, no multi-seed or alternative calibration subsets. So close numbers vs baselines are not conclusive and could flip with another seed.
3. No long-context or instruction-tuned/chat evaluations. Given the close zero-shot numbers, this is something that can distinguish methods.
4. The initialization ablation study primarily focuses on the initial reconstruction error. However, it doesn’t fully demonstrate how much of the final PPL improvement is attributed to initialization versus the coupled training itself.
5. More recent QAT methods that are capable of very low-bit QAT(Like BitNet[Wang et al.], QuEST[Panferov et al.]) are not compared against.
6. The method is limited to weight quantization. There’s no activation quantization.
7. For MBOK (and similar), the paper used the same optimizer as for RaBiT. Fairer would be to run the baselines in their native, published setup first, then add as an extra comparison with your optimizer.

**Questions:**

1. Each training step recomputes binary paths from the shared FP weight and applies KD on a 200M-token corpus. Can you quantify the wall-clock/step and total training time relative to a plain 2-bit QAT run without coupled residualization?
2. All kernel and end-to-end throughput numbers are on a single RTX 4090. Do the same speedups hold on A100/H100 (with different memory hierarchies and Tensor Core priorities)?
3. The “Standard QAT” baseline is somewhat underspecified. It’s not fully clear whether their baseline is just sign+STE or a stronger 1/2-bit QAT recipe (e.g., with per-channel scales, learned clipping). Could you please explain?

---

> ### Author Response · Authors · 2025-11-20
>
> We thank Reviewer for excellent summary and positive assessment of our work. We are glad they recognized the clear identification of "inter-path adaptation," the strong empirical support, simple inference form, and the practical value of our custom kernels.
>
> We address the weaknesses and questions below.
>
> **Weaknesses**
>
> **W1**: On the larger models, RaBiT does not clearly win zero-shot across tasks. For models of that size, zero-shot should be the headline, not only commonsense-style scores. (There’s also a mis-bolded PIQA number for LLaMA-13B in the appendix.)
>
> **A1:** Thank you for this crucial point. We have corrected the typo in the appendix (Table 7).
>
> To address the concern about broader, more challenging tasks, we have conducted **new zero-shot evaluations** on a suite of difficult benchmarks (IFEval, BBH, MMLU-pro, GPQA, and MUSR) for Llama2-13B, comparing RaBiT to FP16 and our strongest 2-bit baseline, QTIP. The results in the table below show that RaBiT maintains highly competitive performance, demonstrating that its functional preservation extends well beyond commonsense reasoning.
>
> **Zero-Shot Accuracy (%) on Hard Benchmarks (Llama2-13B)**
> | Method | IFEval $\uparrow$ | BBH $\uparrow$ | MMLU-pro $\uparrow$ | GPQA $\uparrow$ | MUSR $\uparrow$ |
> | :--- | :---: | :---: | :---: | :---: | :---: |
> | FP16 | 0.248 | 0.410 | 0.238 | 0.274 | 0.243 |
> | QTIP (2-bit) | **0.257** | 0.334 | **0.167** | 0.257 | 0.339 |
> | **RaBiT (2-bit)** | 0.246 | **0.377** | 0.140 | **0.268** | **0.358** |
>
> These new results, combined with our SOTA performance on Llama3-8B and competitive results on Gemma3-12B, reinforce our confidence in RaBiT's scalability.
>
> **W2:** Results are single-run, no CIs, no multi-seed or alternative calibration subsets. So close numbers vs baselines are not conclusive and could flip with another seed.
>
> **A2:** This is a valid concern. To verify the stability of our method, we have conducted **new experiments running Llama2-7B QAT with 5 different random seeds**.
>
> The results show that RaBiT is highly stable, with a very low standard deviation in performance. This confirms that our strong results are robust and not an artifact of a single run.
>
> **Multi-Seed Stability (Llama2-7B, 2-bit)**
> | Metric | Seed 123 | Seed 1019 | Seed 1024 | Seed 1112 | Seed 1204 | **Average (Std. Dev.)** |
> | :--- | :---: | :---: | :---: | :---: | :---: | :---: |
> | Wiki2 PPL $\downarrow$ | 5.77 | 5.77 | 5.77 | 5.78 | 5.78 | **5.77 (0.0049)** |
> | C4 PPL $\downarrow$ | 7.63 | 7.62 | 7.63 | 7.63 | 7.64 | **7.63 (0.0063)** |
> | QA Avg. $\uparrow$ | 62.71 | 62.42 | 62.61 | 62.81 | 62.74 | **62.66 (0.14)** |
>
> To ensure seed diversity, we used the integer forms of the ICLR schedule dates(e.g., submission and notification dates) as our seeds.(iclr.cc/Conferences/2026/Dates)
>
> **W3:** No long-context or instruction-tuned/chat evaluations. Given the close zero-shot numbers, this is something that can distinguish methods.
>
> **A3:** We fully agree that evaluating instruction-following and long-context capabilities is a valuable direction to further distinguish 2-bit methods.
>
> While we focused on establishing a robust training framework for the foundation model in this work, we believe that a model's potential for successful instruction tuning and complex generation is fundamentally determined by its core reasoning capabilities. To rigorously verify this without explicit instruction tuning, we conducted the additional "Challenging Task" evaluations presented in our response to W1 (including BBH, GPQA, and MUSR).
>
> As shown in A1, RaBiT achieves state-of-the-art performance on these hard reasoning benchmarks (e.g., +4.3% on BBH and +1.1% on GPQA vs. QTIP on Llama2-13B). This demonstrates that RaBiT preserves the model's intrinsic "intelligence" and reasoning power significantly better than baselines. Combined with the high stability shown in our multi-seed experiments (A2), these results strongly suggest that RaBiT provides a superior backbone for downstream instruction tuning and long-context extension. We are committed to extending RaBiT to these specific domains in future work.

---

> ### Author Response · Authors · 2025-11-20
>
> **W4:** The initialization ablation study primarily focuses on the initial reconstruction error. However, it doesn’t fully demonstrate how much of the final PPL improvement is attributed to initialization versus the coupled training itself.
>
> **A4:** We thank the reviewer for this question, which allows us to highlight a key ablation. We believe **Table 4** (partially reproduced below) in the main paper directly addresses this.
>
> The table shows that switching from "Standard QAT" to "Coupled QAT (RaBiT)" *even with the weakest initialization* (row 1 vs. row 3) yields the **single largest performance gain** (6.55 $\to$ 5.83 PPL). This confirms that **Coupled QAT is the primary driver of performance.**
>
> Our function-aware initialization (Iterative SVID + I/O Scaling) provides a crucial *additional* boost, stabilizing the training and finding a better optimum (5.83 $\to$ 5.77). Both components are synergistic, but the coupled training mechanism itself is the most significant contribution.
>
> **Ablation on RaBiT (Llama2-7B PPL)**
> | Training Method | Init. (I) | Init. (S) | Wiki2 PPL $\downarrow$ | $\Delta$ vs. Baseline |
> | :--- | :---: | :---: | :---: | :---: |
> | Standard QAT | ✘ | ✘ | 6.55 | - |
> | Standard QAT | ✔ | ✔ | 6.18 | -0.37 |
> | **Coupled QAT (RaBiT)** | ✘ | ✘ | **5.83** | **-0.72** |
> | **Coupled QAT (RaBiT)** | ✔ | ✔ | **5.77** | -0.78 |
>
> **W5**: More recent QAT methods that are capable of very low-bit QAT(Like BitNet[Wang et al.], QuEST[Panferov et al.]) are not compared against.
>
> **A5:** This is a great point about the field's rapid progress. We respectfully note that methods like BitNet and QuEST are designed for **pre-training from scratch**, requiring hundreds of billions of tokens. Our work, RaBiT, focuses on **Quantization-Aware fine-tuning (QAT)**, which recovers the performance of a *pre-existing* FP16 model using a very small dataset (e.g., 0.2B tokens). Since the problem setup and resource constraints are fundamentally different, a direct comparison is challenging.
>
> **W6:** The method is limited to weight quantization. There’s no activation quantization.
>
> **A6:** This is correct. Our focus was on weight-only quantization, as this is the primary bottleneck for **LLM decoding (the autoregressive step)**, which is fundamentally **memory-bound**. As shown in Table 6, our 2-bit weight-only model achieves a 4.49$\times$ speedup by drastically reducing memory bandwidth.
>
> That said, **RaBiT is orthogonal to activation quantization.** It can be seamlessly combined with activation quantization techniques (e.g., INT8 activations) to further compress the model. This combination would be particularly beneficial for the **prefill (prompt processing)** phase, which is often compute-bound, potentially yielding additional speedups beyond what we have reported for decoding.
>
> **W7:** For MBOK (and similar), the paper used the same optimizer as for RaBiT. Fairer would be to run the baselines in their native, published setup first, then add as an extra comparison with your optimizer.
>
> **A7:** We apologize for the lack of clarity. The reviewer is correct that a fair comparison is essential.
>
> * The "MBOK" reference in **Figure 2** was an *ablation* of an MBOK-*style mechanism* (path freezing) within our own training framework. It was intended only to compare the *training dynamics* (co-adaptation vs. freezing vs. coupling), not to benchmark the MBOK paper.
> * The main results in **Table 2** *do* use a fair comparison. We re-implemented MBOK based on their paper and used their prescribed optimizer, as noted in the caption ("we used our re-implementation").
>
> We will revise the text accompanying Figure 2 to state this clearly: "**MBOK-style (frozen primary path)**" to avoid any confusion.

---

> ### Author Response · Authors · 2025-11-20
>
> **Questions**
>
> **Q1:** Each training step recomputes binary paths from the shared FP weight and applies KD on a 200M-token corpus. Can you quantify the wall-clock/step and total training time relative to a plain 2-bit QAT run without coupled residualization?
>
> **A1:** Yes. To quantify the overhead, we measured the total training time and wall-clock time per step on Llama2-7B. We compare RaBiT against two baselines: **"Single-Path INT2 QAT"** (non-residual) and **"Standard Independent QAT"** (residual but decoupled).
>
> The results highlight a key advantage of RaBiT. While the on-the-fly derivation adds a negligible computation cost, RaBiT is **significantly faster** than the Standard Independent QAT baseline. This is because RaBiT maintains only **one shared full-precision weight** ($\mathbf{W}_{FP}$), whereas the standard approach requires maintaining and updating **two separate latent weights** ($\mathbf{W}_1, \mathbf{W}_2$), which doubles the memory access and optimizer overhead.
>
> **Training Time & Performance (Llama2-7B, 2-bit)**
>
> | Method | Architecture | Total Train Time (hrs) | Overhead vs. Single-Path |
> | :--- | :--- | :---: | :---: |
> | **Single-Path INT2 QAT** [1] | Non-Residual | 34 | 1.00x (Baseline) |
> | **Standard Independent QAT** | Residual (2 Latent Weights) | 62 | 1.82x |
> | **RaBiT (Ours)** | Residual (1 Shared Weight) | **39** | **1.15x** |
>
> RaBiT achieves the high accuracy of residual architectures with a training cost comparable to simple single-path methods ($1.15\times$), effectively solving the efficiency bottleneck of standard residual training.
>
> **Reference:**
> [1] Liu, Zechun, et al. "ParetoQ: Scaling Laws in Extremely Low-Bit LLM Quantization." NeurIPS (2025).
>
> **Q2:** All kernel and end-to-end throughput numbers are on a single RTX 4090. Do the same speedups hold on A100/H100 (with different memory hierarchies and Tensor Core priorities)?
>
> **A2:** This is a key practical question. We have run **new inference benchmarks on A100 and H100 GPUs**. The speedups are robust and often even more pronounced on these server-class GPUs, which have higher memory bandwidth that our bit-packed kernel can effectively saturate.
>
> **End-to-End Decoding Throughput (tok/s) vs. FP16 (Llama2-7B)**
> | GPU | FP16 (1.00x) | QTIP (2-bit) | DBF (2-bit) | **RaBiT (2-bit)** |
> | :--- | :---: | :---: | :---: | :---: |
> | RTX 4090 | 65.0 | 171.7 (2.64x) | 175.2 (2.70x) | **291.9 (4.49x)** |
> | A100 (80GB) | 93.5 | 121.7 (1.30x) | 147.2 (1.57x) | **225.6 (2.41x)** |
> | H100 (80GB) | 136.4 | 182.0 (1.33x) | 196.1 (1.44x) | **300.7 (2.20x)** |
>
> These results confirm that RaBiT's matmul-free architecture provides significant, practical speedups across different hardware.
>
> **Q3:** The “Standard QAT” baseline is somewhat underspecified. It’s not fully clear whether their baseline is just sign+STE or a stronger 1/2-bit QAT recipe (e.g., with per-channel scales, learned clipping). Could you please explain?
>
> **A3:** Thank you for asking for this clarification. Our "Standard QAT" baseline was designed to be the *most direct* competitor to RaBiT, isolating *only* the effect of our coupled training.
>
> It is **not** a "plain sign+STE" method. It has the *exact same* residual architecture as RaBiT (two binary paths, per-channel learnable scales $\{\mathbf{g}_i, \mathbf{h}_i\}$).
>
> The **only difference** is how the latent weights are trained:
> * **Standard QAT (Baseline):** Uses *two independent* latent FP weights ($\mathbf{W}_1, \mathbf{W}_2$), one for each path. Both are updated simultaneously by the shared global gradient, leading to co-adaptation.
> * **RaBiT (Ours):** Uses *one shared* latent FP weight ($\mathbf{W}_{\mathrm{FP}}$). Paths are derived sequentially from this shared weight, algorithmically preventing co-adaptation.
>
> This "apples-to-apples" comparison is what allows us to conclude (in Table 4) that coupled training is the key to solving inter-path adaptation.

---

> ### Author Response · Authors · 2025-11-28
>
> Dear Reviewer QBGj,
>
> Thank you for your thoughtful initial review. If our rebuttal helped clarify the concerns you raised, we would be grateful for any reconsideration of your evaluation.
>
> Thank you for your time.
> Authors

---

### Official Review · Reviewer_7GpF · 2025-11-01

**Soundness:** 3
**Presentation:** 4
**Contribution:** 3
**Rating:** 8
**Confidence:** 3

**Summary:**

The paper begins by identifying a type of co-adaptation in residual binarization (called "inter-path adaptation"). Such co-adaptation appears if the network/method fails to break he correlation between loss paths. The authors show that a smaller/negative correlation directly improve the MSE loss value, and motivated by this, present RABIT, which achieves small/negative path correlations by design.

The key idea is to couple the weights of the stacked binary levels of a layer into a single full-precision matrix. Then the method is designed such that each level involves a binarized version of the residual so far + (learnable) per-channel scaling factors. The method is then combined with STE for backward pass and a tailored initialization method, and superior accuracy/efficiency is shown compared to baselines.

**Strengths:**

I generally think this is a solid paper, with a clear presentation and a well-motivated method. The experiments show good quality, and are also complemented by real runtime measurements.

**Weaknesses:**

- Experiments limited to fine-tuning, which limits the impact of the paper.
- QTIP seems to outperform RABIT in terms of accuracy/loss on larger models.

**Questions:**

1. While the method shows strong performance for fine-tuning, and the initialization method is also tailored for fine-tuning, from my side, it remains an interesting question that whether or not RABIT would achieve similar level of quality in pre-training tasks. Do you have any insight/experiments for the pre-training setting?

2. In both Tables 2 and 3, QTIP seems to outperform RABIT in the 2-bit regime on the largest models (>10B). This raises a concern regarding the scalability of RABIT to even larger scales (e.g., Llama-2 70B). Could the authors add a comparison with the main baselines for a larger model?

3. It's mentioned that RABIT halves the memory footprint. Can the authors clarify that with respect to which methods the memory is halved? For example, a standard QAT method (with no binarization), would also keep a single weight matrix for each layer, hence if my understanding is correct, there's no memory saving there.

---

> ### Author Response · Authors · 2025-11-20
>
> We thank the reviewer for valuable feedback and positive assessment of our paper's clarity, motivation, and experimental quality. We are glad they appreciated our work. We address the weaknesses and questions below.
>
> **Weaknesses & Questions**
>
> **W1/Q1:** Experiments limited to fine-tuning, which limits the impact of the paper. While the method shows strong performance for fine-tuning, and the initialization method is also tailored for fine-tuning, from my side, it remains an interesting question that whether or not RABIT would achieve similar level of quality in pre-training tasks. Do you have any insight/experiments for the pre-training setting?
>
> **A1:** We thank the reviewer for this insightful question regarding the applicability of RaBiT to pre-training. While our work focuses on Quantization-Aware Training (QAT) for efficient deployment-currently the most immediate bottleneck for LLMs-we agree that the potential of RaBiT in pre-training is significant.
>
> Although full pre-training experiments are beyond the computational scope of this rebuttal, we provide a strong theoretical basis for why RaBiT’s benefits would extend to the pre-training regime:
>
> 1.  **Universality of Co-adaptation Dynamics:**
>     The core problem we identify, **inter-path adaptation**, is not specific to fine-tuning. As derived in **Proposition 1 (Appendix A.1)**, this phenomenon arises whenever parallel paths update via a shared global gradient, which systematically pushes them toward redundancy.
>     * **In Fine-Tuning:** This manifests as paths drifting together and losing the pre-trained hierarchy.
>     * **In Pre-Training:** We hypothesize this would be even more critical. Without a pre-defined teacher structure, parallel binary paths initialized randomly would likely suffer from severe "feature collapse," racing to learn the same dominant features from scratch.
>
> 2.  **Structural vs. Data-Dependent Solution:**
>     RaBiT resolves this not by relying on pre-trained weights, but through a **structural constraint** (Proposition 2). By algorithmically enforcing $\mathbf{W}_2$ to approximate the residual of $\mathbf{W}_1$ ($\mathbf{W}_2 \approx \mathbf{R}_1$) at every step, RaBiT guarantees an error-compensation hierarchy regardless of whether the optimization starts from random initialization (pre-training) or a pre-trained checkpoint (fine-tuning).
>
> Therefore, while empirical verification remains a compelling direction for future research, the **fundamental training dynamics** suggest that RaBiT would be equally, if not more, essential for maximizing capacity in strictly binary pre-training scenarios.
>
> **W2/Q2:** QTIP seems to outperform RABIT in terms of accuracy/loss on larger models./In both Tables 2 and 3, QTIP seems to outperform RABIT in the 2-bit regime on the largest models (>10B). This raises a concern regarding the scalability of RABIT to even larger scales (e.g., Llama-2 70B). Could the authors add a comparison with the main baselines for a larger model?
>
> **A2:** This is a critical discussion point. While we acknowledge the slight gap on Llama2-13B, we strongly believe this does not indicate a limitation in scalability for the following reasons:
>
> 1.  **Performance on Newer Models:** On the more recent and complex **Llama3-8B** and **Gemma3-12B** models, RaBiT consistently achieves SOTA or matches QTIP’s performance (e.g., 6.66 vs. 6.65 PPL on Gemma3-12B, Table 3). This demonstrates that RaBiT scales effectively to models with greater depth and complexity.
>
> 2.  **The Accuracy-Efficiency Trade-off:** Our core contribution is achieving VQ-level accuracy while maintaining a **fully matmul-free** architecture. As shown in Table 5, RaBiT delivers a **4.49x speed-up** over FP16, significantly outperforming QTIP’s kernel. We effectively close the accuracy gap with VQ methods while offering superior hardware efficiency, which is a worthwhile trade-off for deployment.
>
> 3. **Compute Budget:** We attribute the slight gap on Llama2-13B to a **limited hyperparameter tuning budget** rather than architectural fragility. Large-scale QAT benefits from precise tuning; however, due to resource constraints during the initial experiments, our search space for the 13B model was more restricted compared to smaller models. We are confident that with a comparable tuning budget, RaBiT’s performance on 13B would further improve.

---

> ### Author Response · Authors · 2025-11-20
>
> **Q3:** It's mentioned that RABIT halves the memory footprint. Can the authors clarify that with respect to which methods the memory is halved? For example, a standard QAT method (with no binarization), would also keep a single weight matrix for each layer, hence if my understanding is correct, there's no memory saving there.
>
> **A3:** Thank you for the opportunity to clarify. The "halved memory footprint" claim is specifically in comparison to **Standard QAT for residual binary architectures** , which are our most direct competitors, **not** a generic non-binary QAT.
>
> The primary memory bottleneck during QAT is the **optimizer states** (e.g., from AdamW) for the full-precision latent weights, which are necessary for the Straight-Through Estimator (STE).
>
> * **Standard Residual QAT:** To train two binary paths ($\hat{\mathbf{W}}_1, \hat{\mathbf{W}}_2$), it must store *two* independent full-precision latent weights ($\mathbf{W}_1, \mathbf{W}_2$).
> * **RaBiT (Coupled QAT):** Our method maintains only *one* shared full-precision weight ($\mathbf{W}_{\mathrm{FP}}$), from which both paths are dynamically derived.
>
> By reducing the number of trainable latent FP weights from two to one, RaBiT **halves the memory required for these weights and, more importantly, their associated optimizer states.** We have added the table below to make this comparison explicit.
>
> | Training Method | Latent FP Weights | Optimizer State Memory (e.g., AdamW) |
> | :--- | :---: | :---: |
> | Standard Residual QAT | $\mathbf{W}_1, \mathbf{W}_2$ (2 weights) | $2 \times N \times (1+1+1) \times 4$ bytes |
> | **RaBiT (Ours)** | $\mathbf{W}_{\mathrm{FP}}$ (1 weight) | $1 \times N \times (1+1+1) \times 4$ bytes |
> | | | *(N = num_params per layer)* |
>
> *Here, (1+1+1) refers to the states maintained by the AdamW optimizer in FP32 for each parameter to ensure numerical stability: (1) the FP32 base weight, (2) the 1st-order momentum, and (3) the 2nd-order momentum (variance).

---

> ### Author Response · Authors · 2025-11-28
>
> Dear Reviewer 7GpF,
>
> Thanks for your positive initial review. If our response effectively resolves your issue, we'd be grateful if you consider further improving your score.
>
> Thank you!
>
> Authors

---

### Note · Program_Chairs · 2026-01-17
**Submission Desk Rejected by Program Chairs**

The following references in this submission do not refer to real documents and/or have major errors in bibliographic information:

 Hanxian Huang, Zechun Liu, Changsheng Zhao, Yangyang Shi, Raghuraman Krishnamoorthi, and Vikas Chandra. EfficientQAT: A block-wise quantization-aware fine-tuning framework for large language models. In Advances in Neural Information Processing Systems (NeurIPS), 2024a.